# TRAIN-BEFORE-TEST
# HARMONIZES LANGUAGE MODEL RANKINGS

**Guanhua Zhang, Ricardo Dominguez-Olmedo, Moritz Hardt**
*Max Planck Institute for Intelligent Systems, Tübingen* and *Tübingen AI Center*

## ABSTRACT

Existing language model benchmarks provide contradictory model rankings, even for benchmarks that aim to capture similar skills. This dilemma of conflicting rankings hampers model selection, clouds model comparisons, and adds confusion to a growing ecosystem of competing models. In this paper, we take a different perspective on model comparison: instead of relying on out-of-the-box performance via direct evaluation, we compare *model potential* by providing each model with identical benchmark-specific fine-tuning before evaluation. We call this approach *train-before-test*. Our primary contribution is a comprehensive empirical evaluation of model potential across 24 benchmarks and 61 models. First, we demonstrate that model potential rankings obtained through train-before-test exhibit remarkable consistency across all benchmarks. Whereas traditional rankings demonstrate little external validity under direct evaluation, they enjoy a significant degree of external validity when applying train-before-test: model potential rankings transfer gracefully from one benchmark to another. Second, train-before-test restores the connection between perplexity and downstream task performance, lost under direct evaluation. Remarkably, even pre-finetuning perplexity of a base model predicts post-finetuning downstream performance, suggesting that ranking consistency reflects inherent model potential rather than fine-tuning artifacts. Finally, train-before-test reduces the model-score matrix to essentially rank one, indicating that model potential is dominated by one latent factor, uncovered by train-before-test. While direct evaluation remains useful for assessing deployment-ready performance, train-before-test provides a complementary lens for understanding achievable performance of models after adaptation[†].

## 1 INTRODUCTION

Existing language model benchmarks provide contradictory model rankings, even for benchmarks that aim to capture similar skills (Liang et al., 2023; Beeching et al., 2023; Fourrier et al., 2024). This inconsistency poses a serious challenge: how can we reliably compare, rank, and select models when different benchmarks yield conflicting information? While this ranking disagreement is often attributed to the diverse capabilities of large language models (Ruan et al., 2024), it creates a conundrum in practice that muddles model development decisions (Zhang & Hardt, 2024).

Current evaluation methodology works from *direct evaluation*, probing models via black-box function calls. However, large language models are trained on diverse, often proprietary data mixes that vary significantly across models (Grattafiori et al., 2024; Gemma et al., 2024; Guha et al., 2025). Recent work showed that this leads to the problem of *training on the test task (Dominguez-Olmedo et al., 2024):* the extent to which a model has encountered data similar to the test task during training confounds model comparisons, rankings, and scaling laws (Kaplan et al., 2020). Put simply, an otherwise inferior model may have simply prepared better for a specific task.

In this paper, we take a fresh perspective on evaluation methodology: in contrast with direct evaluation, we compare *model potential* by giving each model the same task-specific fine-tuning. We call this approach *train-before-test*. Its goal is to achieve valid model comparisons by ensuring that all models receive equal preparation for the test.

---

[†]Code is available at `https://github.com/socialfoundations/lm-harmony`.

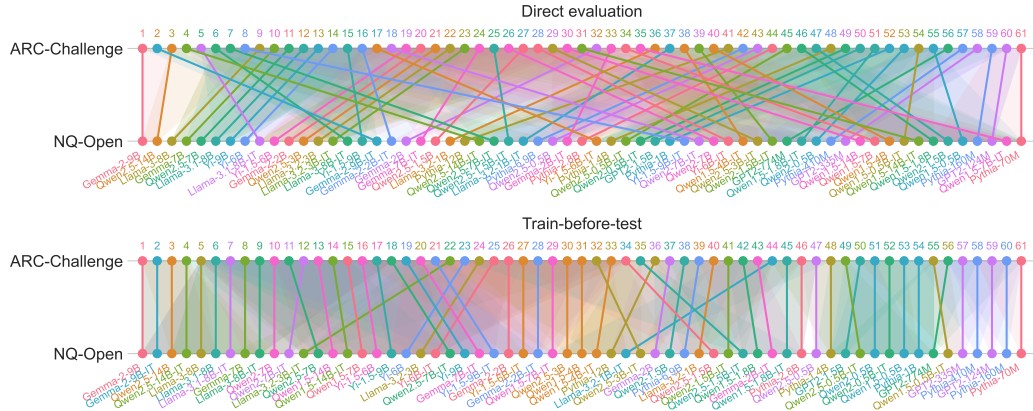

Figure 1: Rankings of 61 language models on two question-answering benchmarks: Natural Questions Open and ARC Challenge. **Top:** Direct evaluation leads to inconsistent rankings. Although both benchmarks test for question-answering ability, the resulting model rankings show substantial disagreement. **Bottom:** Train-before-test aligns model rankings. **Note:** For each of the two plots, we greedily align model rankings as much as possible without violating confidence intervals, thus revealing only those ranking changes that are statistically significant. See Appendix C.1.

We envision train-before-test as a tool for reliable model selection for downstream adaptations. Increasingly, practitioners select one from many available models with the goal of adapting for a specific task. Under direct evaluation the best model to begin with may no longer be the best model after task-specific preparation. In contrast, we show that train-before-task yields model comparisons and rankings that enjoy broad external validity.

## 1.1 OUR CONTRIBUTIONS

**Direct evaluation leads to ranking disagreement even between related tasks.** We demonstrate that the prevalent direct evaluation scheme results in strong disagreement between model ranking across various benchmarks. We show that this strong ranking disagreement persists even when restricting to benchmarks that aim to capture similar tasks. Moreover, rankings still strongly disagree when evaluating models from the same family. The situation presents a serious conundrum for model selection: Under direct evaluation, benchmarks fail to give reliable and actionable insights for model choosing among multiple alternatives.

**Train-before-test leads to consistent model potential rankings.** We comprehensively evaluate train-before-test across 24 benchmark datasets and 61 large language models. By fine-tuning each model on identical task-relevant data before evaluation, we uncover remarkably consistent model potential rankings. Ranking agreement between benchmarks, measured by Kendall's tau, improves for 274 out of 276 benchmark pairs, with the average Kendall's $\tau$ increasing from 0.52 to 0.76. Figure 1 illustrates the result for one typical pair of benchmarks. This consistency suggests that model potential, unlike out-of-the-box performance, has external validity (Salaudeen et al., 2025) and transfers gracefully across different tasks.

**Model potential aligns perplexity rankings with downstream tasks.** Perplexity benchmarks used to be popular, but fell out of favor for public benchmarking and model comparison because of the apparent disconnect between perplexity and downstream task performance (Wei et al., 2022; Ganguli et al., 2022; Liu et al., 2023; Magnusson et al., 2023; Lourie et al., 2025a). We indeed validate this disconnect when comparing model families under direct evaluation. However, train-before-test restores this fundamental relationship in two ways. First, we show that post-fine-tuning perplexity rankings align well with post-fine-tuning downstream task rankings, creating consistency between training objectives and task performance. Second, and more remarkably, for base (non-instruction-tuned) models, even pre-fine-tuning perplexity predicts post-fine-tuning downstream performance.

This suggests that the ranking consistency we observe reflects inherent model potential rather than artifacts of fine-tuning.

**Train-before-test sheds light on the latent factors of benchmark scores.** Consider the large benchmark-model score matrix, where each entry $(i, j)$ corresponds to the performance of model $j$ on a benchmark $i$. Several works have considered this matrix for different reasons and found that it is approximately low rank (Ruan et al., 2024; Owen, 2024; Burnell et al., 2023), but not quite. The first singular value is dominant and correlates with pre-training compute. However, the other components aren't negligible, and their interpretation remains unclear. We show that train-before-test clarifies this state of affairs. After train-before-test, the benchmark-model matrix is essentially rank one. The first principal component accounts for 86% of the explained variance across all models, and for 93% of the variance for a single model family. This suggests that model potential is dominated by a single latent factor, while the additional components observed in direct evaluation may reflect task-specific training exposure.

## 2 RELATED WORK

Benchmarking has played a central role in the advancement of machine learning (Liberman, 2010; Hardt & Recht, 2022). While absolute model performance is often fragile to even seemingly minor changes in evaluation data (Candela et al., 2009; Torralba & Efros, 2011; AlBadawy et al., 2018; Taori et al., 2020; Tsipras et al., 2020), relative model performance—that is, model rankings—tends to transfer surprisingly well across classical benchmarks (Yadav & Bottou, 2019; Recht et al., 2019; Miller et al., 2020). For instance, prior work (Kornblith et al., 2019; Barbu et al., 2019) has shown that model rankings on ImageNet (Deng et al., 2009) also transfer to other image classification and object recognition benchmarks. Moreover, Salaudeen & Hardt (2024) demonstrated that ImageNet rankings remain robust even under major dataset variations. This transferability of model rankings is highly desirable, as it indicates that progress on specific benchmarks reliably reflects broader scientific advancements (Liao et al., 2021; Hardt, 2025).

However, the emergence of foundation models has dramatically transformed the benchmarking landscape compared to the ImageNet era (Liang et al., 2023; Srivastava et al., 2022; Weidinger et al., 2025). With huge training costs and much improved capabilities (Yang et al., 2025; Grattafiori et al., 2024; Ramesh et al., 2021; Gemini, 2023; OpenAI, 2023), practitioners now lean towards directly evaluating LLMs across a wide range of different benchmarks, in the hope of obtaining a more comprehensive assessment of their capabilities (Liang et al., 2023; Suzgun et al., 2022; Hendrycks et al., 2020; Beeching et al., 2023; Fourrier et al., 2024). This shift introduces new challenges, as model rankings across different tasks may vary significantly (Huan et al., 2025; Lourie et al., 2025b). Zhang & Hardt (2024) draw an analogy between multi-task benchmarks and voting systems (Arrow, 1951), revealing that a multi-task benchmarking approach with diverse rankings inherently lacks robustness to minor changes and thus cannot provide a stable unified ranking.

This lack of unified ranking is sometimes seen as a desirable feature within the community (Liang et al., 2023). Some argue that variability reflects the multifaceted strengths and weaknesses of LLMs, suggesting that users should select the best model tailored to their specific needs (Ghosh et al., 2024; Zhang et al., 2024b; Shnitzer et al., 2023). For example, a user who focuses on mathematical tasks could prioritize the math benchmark to choose the optimum model. However, there are two significant concerns regarding this approach: First, the user-driven selection strategy poses challenges for model developers. Given the resource-intensive nature of LLM development (Guo et al., 2025), it is impractical to release a different model for every potential use case. Moreover, developers typically aim to create a general-purpose model (Yang et al., 2025; Grattafiori et al., 2024); however, such a desideratum is often difficult to reliably measure due to the inconsistent rankings observed across benchmarks. Second, we demonstrate in this paper that benchmarks within the same task category can still exhibit substantial discrepancies in model rankings.

One potential reason for the observed inconsistencies in model rankings is that models vary substantially in their training data (Gadre et al., 2023; Albalak et al., 2025). In particular, Dominguez-Olmedo et al. (2024) show that models vary in their degree of preparedness for popular benchmarks, which confounds model evaluations. Inspired by this finding, we investigate a different question: if varying preparedness confounds evaluations, can equalizing preparedness harmonize contradictory model

rankings across benchmarks? We therefore introduce the notion of train-before-test, wherein we fine-tune each model on the corresponding training set so every model arrives well prepared. Through comprehensive evaluations across 24 benchmarks, we demonstrate that train-before-test harmonizes otherwise contradictory model rankings, establishing it as a practical evaluation methodology.

While extensive literature exists on investigating different fine-tuning strategies for LLMs (Zhang et al., 2024a; Zeng et al., 2025; Lester et al., 2021), this lies outside the scope of our investigation. Instead, we apply standardized fine-tuning (Mangrulkar et al., 2022) as an evaluation tool to give all models equivalent preparation before testing. Another relevant area is the literature on scaling laws (Kaplan et al., 2020). Lin et al. (2024) predict full-finetuning performance of a single model from partial finetuning on one task using their rectified scaling law. Zhang et al. (2024a) study how different factors scale within individual tasks. These scaling law approaches model performance changes using model-specific and task-dependent parameters. In contrast, we investigate how standardized fine-tuning harmonizes model rankings across diverse benchmarks in the wild, examining 61 models from six families across 24 tasks spanning multiple categories. Complementary to our focus, Heineman et al. (2025) analyzes the statistical reliability of existing benchmarks, while Gu et al. (2024) proposes a standardized evaluation protocol to reduce variability arising from formatting and scoring choices.

## 3 EXPERIMENTS

### 3.1 EXPERIMENT SETTING

**Benchmark selection.** We begin our study with the `lm-eval-harness` package (Gao et al., 2023), which offers a comprehensive suite of language model benchmarks. To accommodate the train-before-test methodology which requires a dedicated training set for fine-tuning, we first identify benchmarks that provide at least 1,000 training examples. This yields a total of 37 benchmarks, which we broadly categorize into 28 likelihood-based and 9 generation-based benchmarks.

Table 1: We categorize benchmarks into language understanding (LU), commonsense reasoning (CR), question answering (QA), physics/biology/chemistry (PBC), math (Math), and medicine (Med).

| Category | Benchmarks |
|---|---|
| LU | `MNLI`, `QNLI`, `RTE`, `CoLA`, `SST-2`, `MRPC`, `QQP`, `WiC`, `ANLI` |
| CR | `Winogrande`, `CommonsenseQA`, `Hellaswag`, `Social-IQA` |
| QA | `OpenBookQA`, `NQ-Open`, `BoolQ`, `ARC-Easy`, `ARC-Challenge` |
| PBC | `SciQ`, `PIQA` |
| Math | `MathQA`, `GSM8K` |
| Med | `MedMCQA`, `HeadQA` |

Generation-based benchmarks are often computationally intensive to evaluate, as base models typically generate text until reaching their maximum sequence length. These benchmarks are also over-challenging for smaller models with limited parameters, such as GPT-2 (Radford et al., 2019). Therefore, we select only `NQ-Open` and `GSM8K` from the generation-based benchmarks. Among the likelihood-based benchmarks, we further exclude six due to observed anomalies during fine-tuning, such as a lack of performance improvement in over 20% of models. See Appendix A.1 for details.

Our final selection consists of 24 benchmarks covering diverse domains and task types. These benchmarks are primarily multiple-choice question answering benchmarks, with accuracy as the task metric. We categorize all benchmarks by their descriptions, see Table 1. If a benchmark does not come with a validation split, we randomly allocate 20% of the training data as the validation set. To save computational resources, we cap the number of training data at 50,000, validation data at 1,000, and testing data at 10,000.

**Model selection.** We consider 61 language models across six model families: LLAMA (Grattafiori et al., 2024), QWEN (Yang et al., 2025), GEMMA (Gemma et al., 2024), PYTHIA (Biderman et al., 2023), GPT-2 (Radford et al., 2019) and YI (Young et al., 2024). Due to computational constraints, we select models with no more than 14B parameters. See Table 2 for the full list. We include both base and instruction-tuned models, and use the suffix -IT to denote instruction-tuned models.

**Evaluation setup.** We evaluate the 61 models across all 24 benchmarks using both direct evaluation and train-before-test evaluation. We use the `lm-eval-harness` library for evaluation. We evaluate models zero-shot (Brown et al., 2020). For direct evaluation, we simply evaluate the model as it is. For train-before-test, we fine-tune models for five epochs using learning rates in $\{1e-5, 2e-5, 5e-5\}$,

Table 2: Models considered, categorized by model family.

| Family | Model Name Suffix |
|---|---|
| LLAMA- | 3-8B, 3.1-8B, 3.2-1B, 3.2-3B, 3-8B-IT, 3.1-8B-IT, 3.2-1B-IT, 3.2-3B-IT |
| QWEN- | 1.5-0.5B, 1.5-1.8B, 1.5-4B, 1.5-7B, 1.5-14B, 2-0.5B, 2-1.5B, 2-7B, 2.5-0.5B, 2.5-1.5B, 2.5-3B, 2.5-7B, 2.5-14B, 1.5-0.5B-IT, 1.5-1.8B-IT, 1.5-4B-IT, 1.5-7B-IT, 1.5-14B-IT, 2-0.5B-IT, 2-1.5B-IT, 2-7B-IT, 2.5-0.5B-IT, 2.5-1.5B-IT, 2.5-3B-IT, 2.5-7B-IT, 2.5-14B-IT |
| GEMMA- | 2B, 7B, 2-2B, 2-9B, 2B-IT, 7B-IT, 2-2B-IT, 2-9B-IT |
| GPT2- | 124M, 335M, 774M, 1.5B |
| PYTHIA- | 70M, 160M, 410M, 1B, 1.4B, 2.8B, 6.9B, 12B |
| YI- | 6B, 9B, 6B-IT, 1.5-6B, 1.5-9B, 1.5-6B-IT, 1.5-9B-IT |

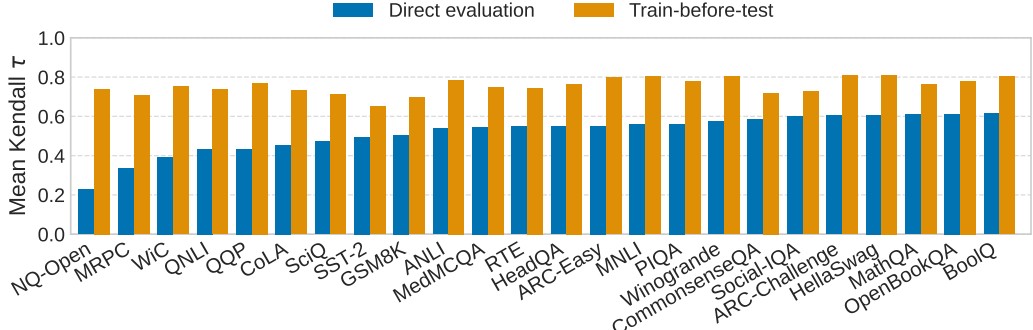

Figure 2: Mean ranking agreement between each benchmark and all others. We calculate Kendall's $\tau$ between each benchmark and every other benchmark, and then average the results. **Compared to direct evaluation, train-before-test consistently improves ranking agreement.** A detailed comparison of Kendall's $\tau$ values for every benchmark pair is provided in Appendix B.1. On average, the overall average Kendall's $\tau$ is 0.52 for direct evaluation and 0.76 for train-before-test.

separately. The best performing checkpoint is then selected based on performance on a separate validation set, yielding $61 \times 24 = 1,464$ fine-tuned models in total. We use parameter-efficient fine-tuning (PEFT) (Hu et al., 2021; Mangrulkar et al., 2022). See more details in Appendix A.2. Each fine-tuned model is then evaluated on the corresponding benchmark's test set. For each benchmark, we rank models according to their performance. We then measure the ranking correlation across pairs of benchmarks using Kendall's $\tau$ (Kendall, 1938).

## 3.2 DOWNSTREAM RANKING AGREEMENT

As depicted in Figure 2, direct evaluation shows only modest ranking agreement between the 24 benchmarks, with an average Kendall's $\tau$ of 0.52. This lack of agreement across benchmarks complicates model assessment and makes it challenging to aggregate results into a meaningful overall ranking (Zhang & Hardt, 2024). In contrast, the train-before-test methodology leads to a substantial improvement in ranking agreement. Under this approach, 274 out of 276 benchmark pairs show higher Kendall's $\tau$ scores, with the average $\tau$ rising from 0.52 to 0.76. This stronger consistency suggests that model potential ranking on one benchmark is likely to generalize to others, including practitioners' own cases, which simplifies model comparison and selection. Notably, benchmarks that appeared to be outliers under direct evaluation, such as NQ-Open and MRPC, demonstrate much greater ranking consistency under train-before-test. For example, the average Kendall's $\tau$ between NQ-Open and all other benchmarks improves from 0.23 to 0.74.

We further split all benchmarks into six categories (e.g., language understanding, math), see Table 1. For each category pair, we report in Figure 3 the intra-category average ranking correlations and inter-category average ranking correlations across all relevant benchmark pairs. Consistent with our previous findings, we observe reasonably poor ranking agreements across categories under direct evaluation. While one might expect high intra-category agreement—after all, tasks within the same category tend to be relatively similar—direct evaluation results in low intra-category agreement in

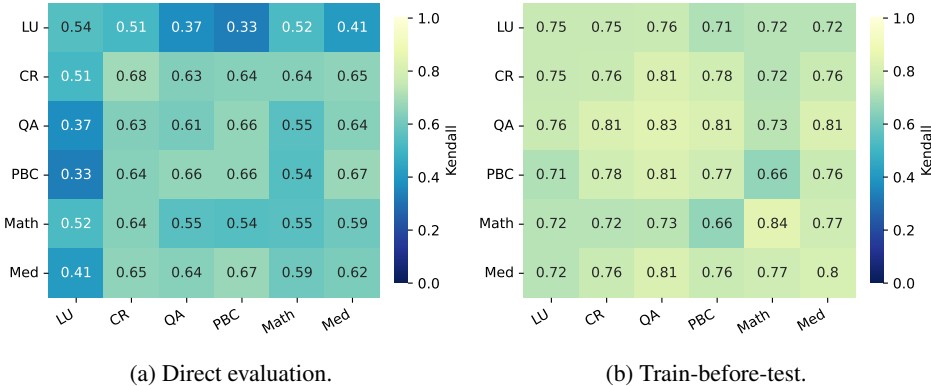

(a) Direct evaluation.

(b) Train-before-test.

Figure 3: Cross-category ranking agreement for direct evaluation (left) and train-before-test (right). We categorize benchmarks into language understanding (LU), commonsense reasoning (CR), question answering (QA), physics/biology/chemistry (PBC), math (Math), and medicine (Med), see Table 1. Kendall's $\tau$ is averaged across all pairs of benchmarks that belong to two given categories. The diagonal entries represent intra-category agreement and the others represent inter-category agreement. **Train-before-test improves both intra- and inter-category ranking agreement in all instances.**

many cases. For example, the intra-category mean Kendall's $\tau$ is 0.54 for language understanding and 0.55 for math. This further underscores the difficulty of selecting models based on direct evaluation. Even if the goal is to choose a model that excels within a specific domain, the low intra-category agreement makes this decision challenging.

In contrast, train-before-test boosts both intra- and inter-category consistency. For example, the intra-category mean Kendall's $\tau$ for language understanding raises from 0.52 to 0.75, as well as from 0.55 to 0.84 for the math category. Moreover, agreement between categories is often nearly as high as agreement within categories. This suggests that models with higher potential in one domain tend to also perform well across other domains after adaptation.

## 3.3 PERPLEXITY AGREEMENT

We now study the agreement between downstream benchmark rankings and perplexity rankings on general domain corpora. To do so, we collect three corpora from `Wiki`pedia, `StackExchange`, and `arXiv`, retaining only contents from 2025. Because all models used were released before 2025, they could not have seen these texts during pretraining. Specifically, we collect 3,366 documents for `Wiki`, 6,001 for `StackExchange` and 44,384 for `arXiv`. These datasets are split into training, validation, and testing sets, in an 8:1:1 ratio.

We measure perplexity in bits per byte, using the `lm-eval-harness` library. We then compute models rankings based on the perplexity evaluations, and compare the rankings with those of the downstream benchmarks considered in earlier sections. We exclude the four GEMMA models from these results, as `lm-eval-harness` provides unreliable perplexity measurements for GEMMA models due to its rolling window implementation. See Appendix B.2 for details.

The results are presented in Figure 4. In contrast to downstream tasks, perplexity rankings demonstrate strong agreement both under direct evaluation and train-before-test. Specifically, the average Kendall's $\tau$ between the perplexity rankings is 0.76 for direct evaluation and 0.78 for train-before-test. We hypothesize that this reasonably strong agreement arises due to the smooth relationship between perplexity evaluations (Brandfonbrener et al., 2024; Mayilvahanan et al., 2025).

When comparing ranking agreement between perplexity evaluations and downstream benchmarks, we find that agreement is low under direct evaluation, with a mean Kendall's $\tau$ of 0.48. This lack of agreement is concerning, as it signals a disconnect between the language modelling pre-training objective and downstream benchmark performance. Fortunately, we find that our train-before-test methodology improves ranking agreement substantially, with the mean Kendall's $\tau$ ranking correlation between perplexity rankings and benchmark rankings rising to 0.74. This finding is reassuring: a

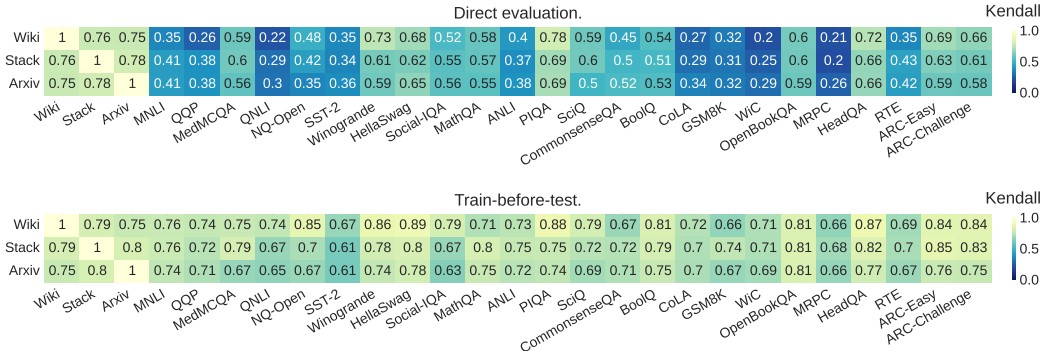

Figure 4: Ranking agreement between perplexity rankings and downstream benchmark rankings under direct evaluation (top) and train-before-test (bottom). Perplexity rankings are consistent with each other under both evaluation schemes, with an average Kendall's $\tau$ of 0.76 and 0.78, respectively. However, for direct evaluation, agreement between perplexity rankings and downstream rankings is low, with an average Kendall's $\tau$ of just 0.48. Fortunately, **train-before-test results in higher agreement between perplexity and downstream**, increasing average Kendall's $\tau$ to 0.74.

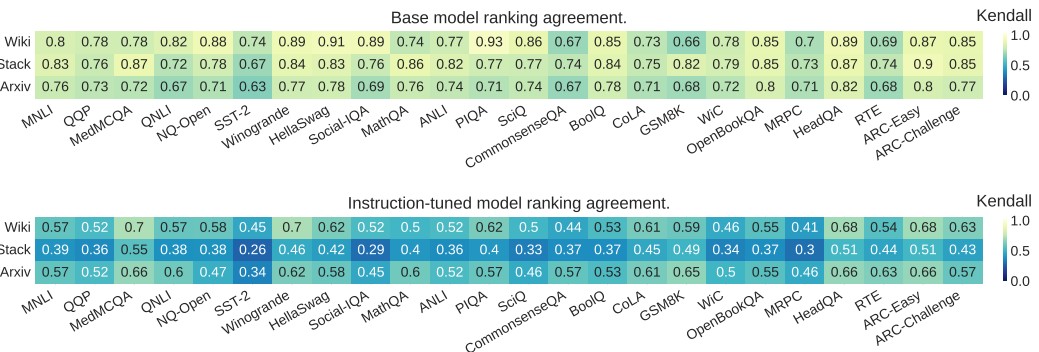

Figure 5: Ranking agreement between perplexity rankings **before fine-tuning** (direct evaluation) and downstream benchmark rankings **after fine-tuning** (train-before-test) for base models (top) and instruction-tuned models (bottom). Unlike Figure 4 where both rankings in each comparison use the same evaluation scheme, here we test whether pre-fine-tuning perplexity can predict post-fine-tuning downstream performance. Base models show strong correlation (average Kendall's $\tau$ = 0.78), suggesting perplexity is a good predictor of model potential. This indicates that the **ranking consistency we observe reflects inherent model potential rather than artifacts introduced by fine-tuning.** Instruction-tuned models show much weaker correlation (average Kendall's $\tau$ = 0.51).

light amount of fine-tuning on task data is sufficient to align the language modeling training objective with downstream performance. Moreover, we find that ranking agreement between perplexity and downstream evaluations is roughly similar to agreement across downstream evaluations. This suggests that, despite perplexity typically not being used for benchmarking purposes, it can be as effective a ranking metric as benchmark evaluations.

Drawing inspiration from prior work (Liu et al., 2023; Xia et al., 2023; Gadre et al., 2024; Du et al., 2024; Zhang et al., 2024a), we further examine the correlation between model rankings according to *average* perplexity across the three text corpora and *average* downstream performance across the 24 benchmarks. Gadre et al. (2024) show that, when models are trained on the same pretraining data, perplexity is well-correlated with aggregate benchmark performance. Our setup differs in that we consider a diverse set of model families, each trained on different pretraining data. Under direct evaluation, we find that the ranking correlation is modest, with a Kendall's $\tau$ of only 0.55. We hypothesize that this relatively weak agreement is due to differences in pretraining data and instruction tuning, resulting in varying levels of exposure to benchmark tasks during training (Dominguez-

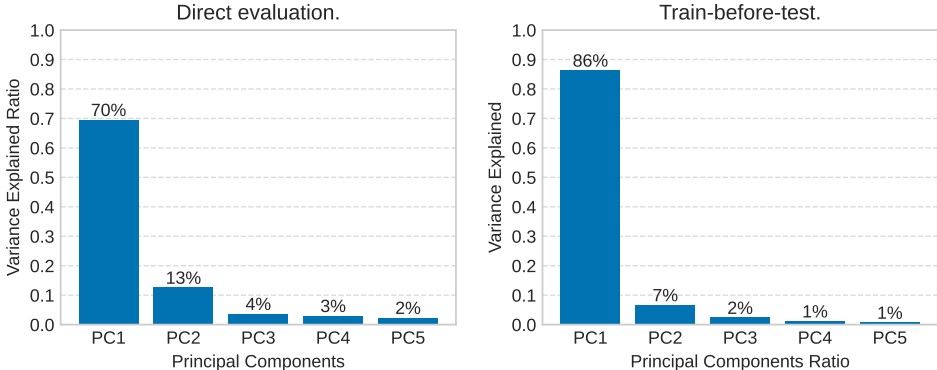

Figure 6: Explained variance ratios of the top five principal components of the benchmark score matrix, under direct evaluation (left) and train-before-test (right). Train-before-test substantially increases the amount of variance explained by the first principal component, from 70% to 86%. **This indicates the model potential is dominated by one single latent factor.**

Olmedo et al., 2024). Fortunately, when applying our train-before-test methodology, the ranking correlation between average perplexity and average downstream performance improves substantially, with Kendall's $\tau$ increasing from 0.55 to 0.84.

We additionally examine the agreement between perplexity prior to fine-tuning and downstream task performance after fine-tuning. That is, between direct evaluation perplexity rankings and train-before-test downstream performance rankings. We plot such ranking agreement in Figure 5, dividing models into base models and instruction-tuned models. For base models, perplexity prior to fine-tuning is a strong indicator of model potential on downstream tasks, with an average Kendall's $\tau$ of 0.78. This indicates that, for base models, direct evaluation of perplexity is already a reasonably reliable metric for ranking models. Moreover, it indicates that the ranking consistency we observe reflects inherent model potential rather than artifacts introduced by fine-tuning.

However, the same does not hold for instruction-tuned models (average Kendall's $\tau = 0.51$). Instruction-tuning renders perplexity rankings unreliable, as ranking agreement is low across the board. This is to be expected: instruction fine-tuning tends to increase both benchmark performance ($\uparrow$) and perplexity ($\downarrow$) on general text corpora, thus clouding the relationship between perplexity and downstream evaluations. Fortunately, as shown earlier, train-before-test restores high ranking agreement between perplexity evaluations and downstream performance.

### 3.4 LOW-RANKED MODEL SCORE MATRIX

So far, we have shown that comparing model potential using the train-before-test yields consistent rankings across benchmarks. We now examine the implications of this finding by analyzing the resulting matrix of model scores, where each entry $(i, j)$ corresponds to the performance of model $j$ on a benchmark $i$. We use Principal Component Analysis (PCA) to examine the structure of the model score matrix.

Figure 6 shows the explained variance ratios of the first five principal components. These results support previous findings that the score matrix is of low rank (Ruan et al., 2024). Under direct evaluation, the first five components account for 91% of the total variance. A similar trend is observed for train-before-test scores, where the first five components explain 97% of the variance. Notably, under train-before-test, the first principal

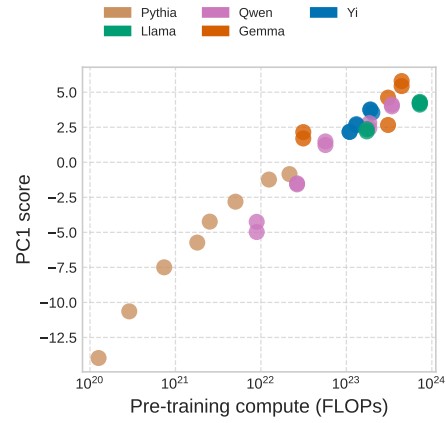

Figure 7: PC1 scores under train-before-test align with the pre-training compute.

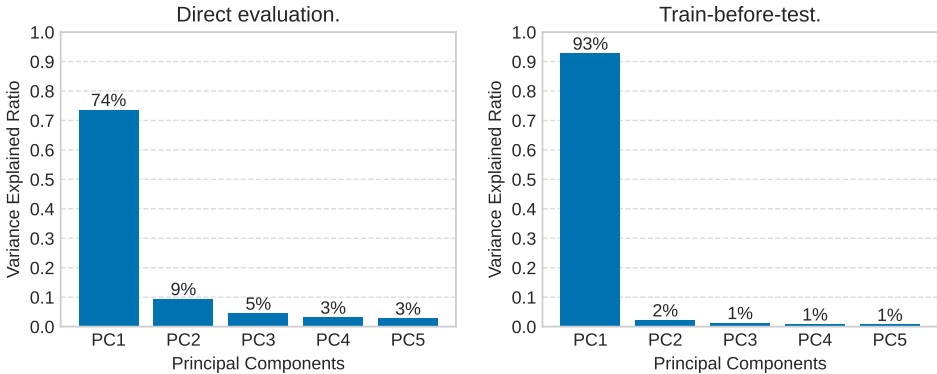

Figure 8: Explained variance ratios of the top five principal components of the QWEN score matrix. For train-before-test, the explained variance ratio of PC1 increases to 93%. **Controlling the model family to QWEN has made the score matrix essentially rank one.**

component (PC1) captures a much larger share of the variance: 86%, compared to 70% for direct evaluation.

Prior works interpret PC1 scores under direct evaluation as an indication of general capability, with later principal components denoting domain-specific capabilities not captured by PC1 (e.g., reasoning ability, coding ability) (Ruan et al., 2024; Burnell et al., 2023). Unlike out-of-the-box performance, which is controlled by multiple factors (Ruan et al., 2024; Burnell et al., 2023), model potential is dominated by one single principal axis. This dramatic change indicates that train-before-test removes confounding factors, such as differential exposure to benchmark-related data during pretraining, that create artificial diversity in rankings. It is of no surprise that PC1 also positively correlates with pre-training compute, as shown in Figure 7[1], which have been identified as crucial to model performances (Kaplan et al., 2020; Hoffmann et al., 2022). See detailed PC1 scores in Figure 11.

**Case study for Qwen models.** We repeat our PCA analysis on the score matrix containing only QWEN models, depicted in Figure 8 (see other models in Appendix B.7). Remarkably, we find that PC1 for train-before-test explains 93% of the variance, roughly as much as the variance explained by the top five principal components under direct evaluation. That is, whereas for direct evaluation the score matrix is low-rank; train-before-test renders the score matrix essentially rank one.

## 4 DISCUSSION, LIMITATIONS, AND CONCLUSION

Train-before-test fundamentally reframes how we interpret model evaluation. Whereas direct evaluation yields benchmark-specific rankings that often contradict one another, train-before-test harmonizes rankings across a wide array of tasks and datasets. This shift from measuring out-of-the-box *performance* to comparing achievable *potential* equips the community with a more stable and externally valid evaluation methodology.

This emphasis on model potential is particularly valuable for scenarios involving model development and adaptation. Practitioners frequently need to make decisions during model development—selecting checkpoints mid–pre-training or choosing a base model for further instruction tuning or domain-specific adaptation. In these scenarios, direct evaluation, while useful for assessing deployment readiness, is of limited relevance and utility. A model that performs poorly on direct evaluation might excel when adapted to new tasks. Train-before-test, by contrast, shows that rankings on any task will also generalize to others, offering more promising guidance for model selection.

One might argue that ranking consistency is unnecessary if we can simply choose benchmarks close to a given downstream application. However, our findings highlight three challenges with that view. First, even benchmarks that purport to measure the same skill (*e.g.*, question answering) produce

---

[1]We only plot models whose number of training tokens is publicly available. See Table 4 for details.

contradictory rankings under direct evaluation. Second, no benchmark perfectly captures the specifics of an application, making some degree of generalization unavoidable. Third, in real deployments, models are often adapted to varying degrees, making the *potential* the relevant signal for comparison.

**Limitations.** Train-before-test requires fine-tuning models on task-specific data before evaluation, which certainly increases the evaluation cost. However, this investment yields dividends through improved reliability. Our findings suggest that fewer benchmarks suffice under train-before-test, as rankings from one benchmark reliably transfer to others. This reduction in required evaluations can offset the per-benchmark cost increase. Second, despite significant improvements in cross-benchmark ranking consistency, the ranking agreements remain imperfect. The residual imperfect correlation may arise from incomplete adaptation with PEFT (Mangrulkar et al., 2022) or irreducible measurement noise (Fisher & Sen, 1994; Heineman et al., 2025). Third, unfortunately, many benchmarks no longer come with training data, making it more difficult to apply train-before-test. In light of our findings, we recommend that future benchmarks provide fine-tuning data for the benchmark. A fourth limitation is that some commercial model providers do not easily allow fine-tuning of their models. We contend that in this case the problem is with the model provider. There is clearly scientific value in creating an ecosystem of models that can be fine-tuned. Train-before-test evaluation creates additional incentives for making models easy to fine-tune.

**Conclusion.** Overall, train-before-test complements existing evaluation practices by distinguishing between *performance* and *potential*. Importantly, potential comparison is not intended to replace direct evaluation—both serve distinct purposes. Direct evaluation remains useful for understanding immediate deployment readiness, while potential comparison provides insights into adaptability and development prospects. Together, they offer a more complete picture of model capabilities. We believe that adopting train-before-test as a standard alongside direct evaluation can significantly improve the reliability, interpretability, and practical utility of the model evaluation ecosystem.

## ACKNOWLEDGEMENT

We thank Yatong Chen, Florian Dorner, Mina Remeli and Jiduan Wu for helpful discussions and feedback on draft versions of this work.

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

# A  ADDITIONAL EXPERIMENT SETTING

## A.1  BENCHMARK SELECTION

We begin our study with the `lm-eval-harness` package (Gao et al., 2023), which offers a comprehensive suite of language model benchmarks. To accommodate the train-before-test methodology which requires a dedicated training set for fine-tuning, we first identify benchmarks that provide at least 1,000 training examples. This yields a total of 37 benchmarks.

These benchmarks can be broadly categorized as 28 likelihood-based and 9 generation-based benchmarks. Likelihood-based evaluations test for the likelihood of different completions given some input string; for example, different answer choices given a multiple-choice input question. Since the number of completions is usually small, likelihood-based evaluations are generally compute-efficient.

Generation-based evaluations, in contrast, generate some output response given an input query. If responses tend to be long, then generation-based evaluations naturally become compute-intensive. This is particularly true for base models, which are usually not trained for instruction following and therefore continue to generate tokens until hitting their maximum token limit. These generation-based benchmarks are also over-challenging for smaller models with limited parameters, such as GPT-2 (Radford et al., 2019). Therefore, we exclude seven generation-based benchmarks, `Drop`, `CoQa`, `ReCoRD`, `bAbi`, `WebQA`, `TriviaQA` and `Fld-Default`. Nevertheless, we retain two widely used generation-based benchmarks, `GSM8K` and `NQ-Open`, in our experiments.

We additionally excluded five benchmarks due to anomalies observed during fine-tuning: `MedQA-4Options`, `LogiQA`, `Mutual`, `Mela-EN`, and `Swag`. For these benchmarks, more than 20% of models, most of which are small models with fewer than 0.5B parameters, showed no performance improvement after fine-tuning. We also excluded `Paws-EN`, as its corresponding model ranking under direct evaluation was negatively correlated (Kendall's $\tau$ less than zero) with 23 out of 24 other benchmarks. We attribute this anomaly to the unusual prompting template used by `lm-eval-harness`.

Our final selection includes 24 benchmarks: `MNLI` (Williams et al., 2018), `QNLI` (Rajpurkar et al., 2016), `RTE` (Dagan et al., 2006; Giampiccolo et al., 2007; Bentivogli et al., 2009), `CoLA` (Warstadt et al., 2018), `SST-2` (Socher et al., 2013), `MRPC` (Dolan & Brockett, 2005), `QQP`, `WiC` (Pilehvar & Camacho-Collados, 2018), `ANLI` (Nie et al., 2020), `Winogrande` (Levesque et al., 2011), `CommonsenseQA` (Talmor et al., 2019), `Hellaswag` (Zellers et al., 2019), `Social-IQA` (Sap et al., 2019), `OpenBookQA` (Mihaylov et al., 2018), `NQ-Open` (Kwiatkowski et al., 2019), `BoolQ` (Clark et al., 2019), `ARC-Easy`, `ARC-Challenge` (Clark et al., 2018), `SciQ` (Welbl et al., 2017), `PIQA` (Bisk et al., 2019), `MathQA` (Amini et al., 2019), `GSM8K` (Cobbe et al., 2021), `MedMCQA` (Pal et al., 2022), `HeadQA` (Vilares & Gómez-Rodríguez, 2019).

## A.2  EVALUATION SETUP

For our train-before-test evaluations, we fine-tune each model for five epochs and select the best-performing checkpoint based on evaluations on a separate validation set. We use the AdamW optimizer with a weight decay of 0.01. For each model-benchmark combination, we perform a hyperparameter search over three learning rates $\{1e-5, 2e-5, 5e-5\}$ and select the optimal one based on validation performance. To reduce memory consumption, we employ parameter-efficient fine-tuning (PEFT) (Hu et al., 2021; Mangrulkar et al., 2022), We use a LoRA configuration with rank 8, $\alpha = 32$, and dropout 0.1. Most of our experiments are conducted on Quadro RTX 6000, Tesla V100-SXM2-32GB and NVIDIA A100-SXM4-80GB GPUs.

In cases where models show no performance improvement after fine-tuning, we report their pre-fine-tuning results. This scenario is rare and typically occur with smaller models (less than 500M parameters) that lack the capacity to perform certain tasks, resulting in near-random performance both before and after fine-tuning. Additionally, since all training datasets in our study are publicly available, some models may have encountered this data during pre-training, potentially limiting the benefits of additional fine-tuning.

For instruction-tuned models, we evaluate performance both with and without chat templates, selecting the configuration that yields better results. Specifically, during direct evaluation, we assess model

performance on the validation set under both conditions and apply the better-performing configuration to the test set. In the train-before-test setting, we similarly fine-tune two variants: one with training data formatted using chat templates and one without. We then select the approach that achieves the best performance on the validation set for final evaluation.

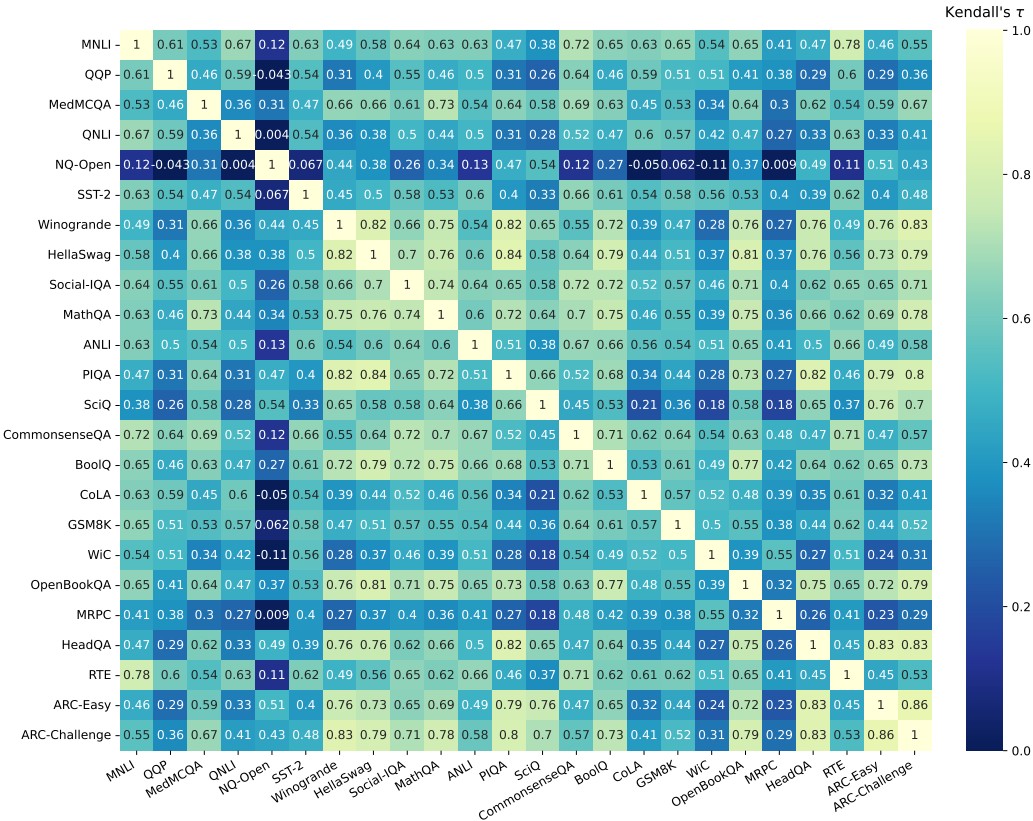

Figure 9: Cross benchmark ranking agreement under direct evaluation. Benchmarks are sorted based on the training dataset size. Kendall's $\tau$ is calculated for every benchmark pair.

# B  ADDITIONAL EXPERIMENT RESULTS

## B.1  DOWNSTREAM RANKING AGREEMENT

We plot detailed pairwise ranking correlation agreement between benchmarks in Figure 9 (direct evaluation) and 10 (train-before-test), corresponding to Figure 2 in the main text.

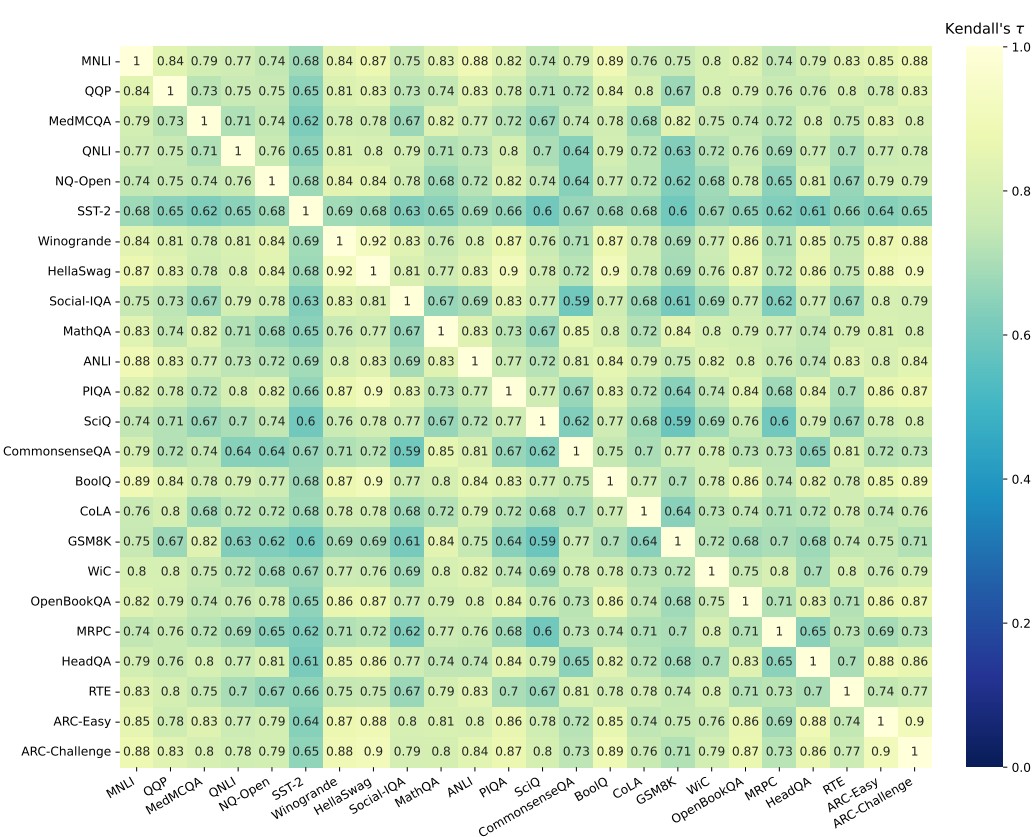

Figure 10: Cross benchmark ranking agreement under train-before-test. Benchmarks are sorted based on the training dataset size. Kendall's $\tau$ is calculated for every benchmark pair.

Table 3: Bits per byte (BPB) of eight excluded GEMMA models compared to PYTHIA-410M across the three newly collected corpora. The GEMMA models exhibit abnormally high BPB values on `Wiki` and `Stack`, likely due to the greater average sequence length in these two datasets. Specifically, `Arxiv` has an average of 163 words per document, compared to 250 for `Stack` and 1502 for `Wiki`.

|  | Arxiv | Wiki | Stack |
|---|---|---|---|
| GEMMA-2B | 0.766 | 1.578 | 1.139 |
| GEMMA-2B-IT | 0.770 | 1.524 | 1.222 |
| GEMMA-7B | 1.013 | 4.780 | 4.053 |
| GEMMA-7B-IT | 1.053 | 18.711 | 20.958 |
| GEMMA-2-2B | 0.730 | 1.784 | 1.340 |
| GEMMA-2-2B-IT | 0.705 | 1.191 | 0.997 |
| GEMMA-2-9B | 0.709 | 2.216 | 1.685 |
| GEMMA-2-9B-IT | 0.638 | 1.234 | 0.978 |
| PYTHIA-410M | 0.791 | 1.065 | 0.945 |

## B.2 PERPLEXITY RANKING AGREEMENT

In this work, we collect three corpora from `Wikipedia`, `StackExchange`, and `arXiv`. We only collect documents from 2025. More specifically, we collect 3,366 documents for `Wiki`, 6,001 for `StackExchange` and 44,384 documents for `arXiv`. These datasets are split into training, validation, and testing sets, in an 8:1:1 ratio. For `arXiv`, we utilize only the paper abstracts, while for `StackExchange`, we use only the questions. Consequently, the average document length is 163 words for `arXiv`, 250 words for `StackExchange`, and 1,502 words for `Wikipedia`.

We exclude GEMMA models from our perplexity agreement experiments, as `lm-eval-harness` provides unreliable perplexity measurements for GEMMA models[2]. We report the bits per byte (BPB) for the GEMMA models in Table 3. While the BPB values for GEMMA on `arXiv` (the dataset with the shortest average sequence length) are mostly reasonable, the performance on `StackExchange` and `Wikipedia` is notably worse, even compared to smaller models like PYTHIA-410M.

This anomaly stems from how `lm-eval-harness` handles long sequences via a rolling window mechanism. Unlike other models, GEMMA requires every input sequence to begin with the `BOS` token. If this constraint is not met, perplexity degrades significantly. Consequently, when processing long sequences that are chunked into multiple windows, GEMMA's performance degrades.

## B.3 PC1 SCORE UNDER TRAIN-BEFORE-TEST

We plot the PC1 scores under train-before-test in Figure 11. We also provide the pre-training compute details for models with publicly available training token counts, as shown in Table 4.

---

[2]See discussion at `https://github.com/huggingface/transformers/issues/29250`.

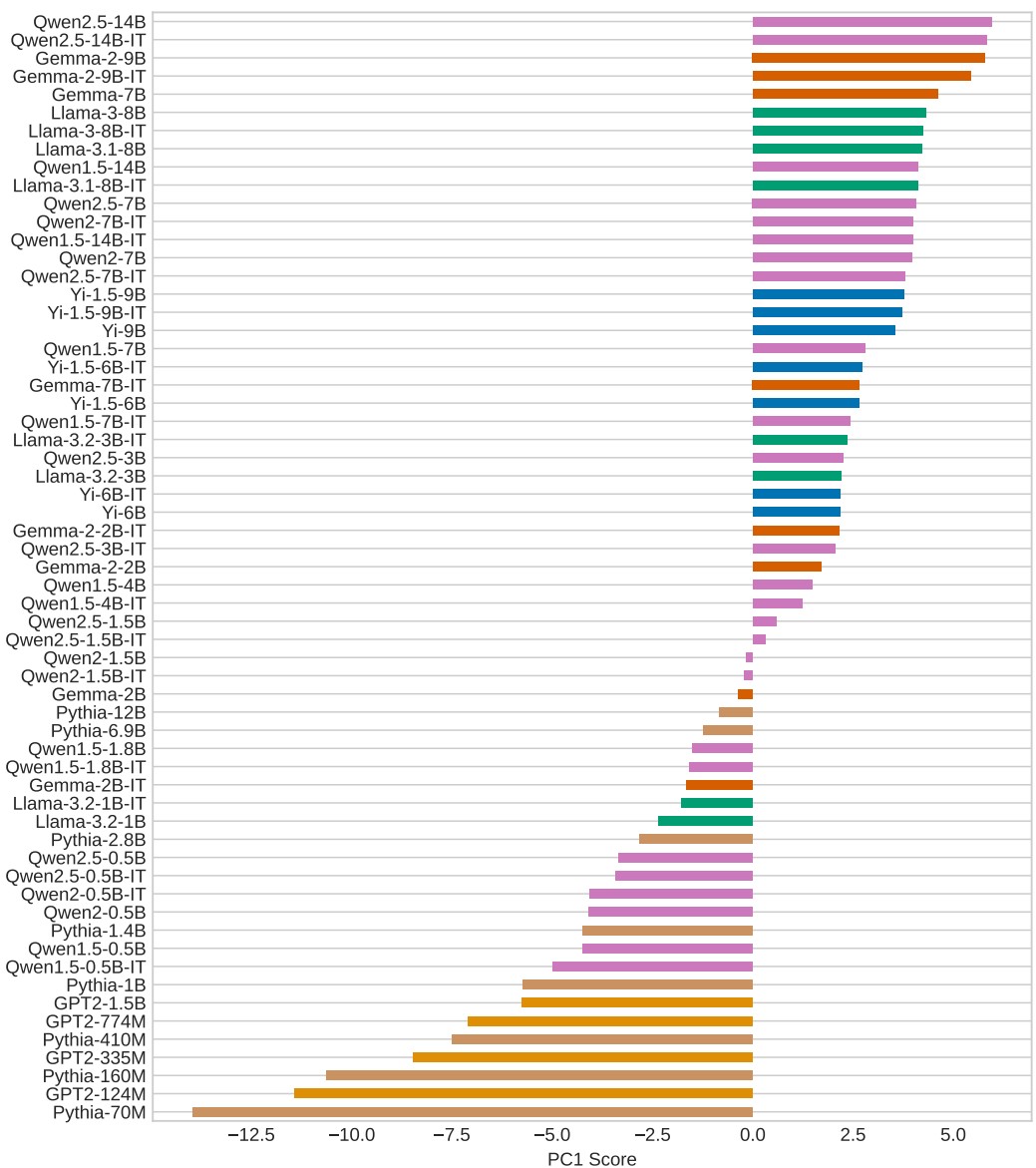

Figure 11: PC1 scores under train-before-test.

Table 4: The models used in Figure 7. The number of training tokens of these models is publicly available. We compute the number of pre-training FLOPs as $6 \times$ #Parameters $\times$ #Tokens.

| Model | #Parameters (B) | #Tokens (B) | #FLOPs ($10^{18}$) |
|---|---|---|---|
| Llama-3-8B | 8.03 | 15000.0 | 722700.00 |
| Llama-3-8B-IT | 8.03 | 15000.0 | 722700.00 |
| Llama-3.1-8B | 8.03 | 15000.0 | 722700.00 |
| Llama-3.1-8B-IT | 8.03 | 15000.0 | 722700.00 |
| Llama-3.2-3B | 3.21 | 9000.0 | 173340.00 |
| Llama-3.2-3B-IT | 3.21 | 9000.0 | 173340.00 |
| Qwen1.5-0.5B | 0.62 | 2400.0 | 8928.00 |
| Qwen1.5-1.8B | 1.84 | 2400.0 | 26496.00 |
| Qwen1.5-4B | 3.95 | 2400.0 | 56880.00 |
| Qwen1.5-7B | 7.72 | 4000.0 | 185280.00 |
| Qwen1.5-14B | 14.20 | 4000.0 | 340800.00 |
| Qwen1.5-0.5B-IT | 0.62 | 2400.0 | 8928.00 |
| Qwen1.5-1.8B-IT | 1.84 | 2400.0 | 26496.00 |
| Qwen1.5-4B-IT | 3.95 | 2400.0 | 56880.00 |
| Qwen1.5-7B-IT | 7.72 | 4000.0 | 185280.00 |
| Qwen1.5-14B-IT | 14.20 | 4000.0 | 340800.00 |
| Gemma-7B | 8.54 | 6000.0 | 307440.00 |
| Gemma-7B-IT | 8.54 | 6000.0 | 307440.00 |
| Gemma-2-2B | 2.61 | 2000.0 | 31320.00 |
| Gemma-2-2B-IT | 2.61 | 2000.0 | 31320.00 |
| Gemma-2-9B | 9.24 | 8000.0 | 443520.00 |
| Gemma-2-9B-IT | 9.24 | 8000.0 | 443520.00 |
| Pythia-70M | 0.07 | 300.0 | 126.00 |
| Pythia-160M | 0.16 | 300.0 | 288.00 |
| Pythia-410M | 0.41 | 300.0 | 738.00 |
| Pythia-1B | 1.00 | 300.0 | 1800.00 |
| Pythia-1.4B | 1.40 | 300.0 | 2520.00 |
| Pythia-2.8B | 2.80 | 300.0 | 5040.00 |
| Pythia-6.9B | 6.90 | 300.0 | 12420.00 |
| Pythia-12B | 12.00 | 300.0 | 21600.00 |
| Yi-6B | 6.06 | 3000.0 | 109080.00 |
| Yi-6B-IT | 6.06 | 3000.0 | 109080.00 |
| Yi-9B | 8.83 | 3800.0 | 201324.00 |
| Yi-1.5-6B | 6.06 | 3600.0 | 130896.00 |
| Yi-1.5-6B-IT | 6.06 | 3600.0 | 130896.00 |
| Yi-1.5-9B | 8.83 | 3600.0 | 190728.00 |
| Yi-1.5-9B-IT | 8.83 | 3600.0 | 190728.00 |

Table 5: We calculate Kendall's $\tau$ between each benchmark and every other benchmark, and then average the results.

|  | Direct evaluation (0 shot) | Direct evaluation (5 shot) | Train-before-test |
|---|---|---|---|
| NQ-Open | 0.23 | 0.44 | 0.74 |
| MRPC | 0.34 | 0.39 | 0.71 |
| WiC | 0.39 | 0.58 | 0.75 |
| QNLI | 0.43 | 0.58 | 0.74 |
| QQP | 0.43 | 0.62 | 0.77 |
| CoLA | 0.45 | 0.65 | 0.73 |
| SciQ | 0.47 | 0.49 | 0.71 |
| SST-2 | 0.50 | 0.62 | 0.65 |
| GSM8K | 0.51 | 0.62 | 0.70 |
| ANLI | 0.54 | 0.56 | 0.78 |
| MedMCQA | 0.55 | 0.66 | 0.75 |
| RTE | 0.55 | 0.59 | 0.74 |
| HeadQA | 0.55 | 0.62 | 0.77 |
| ARC-Easy | 0.55 | 0.63 | 0.80 |
| MNLI | 0.56 | 0.63 | 0.80 |
| PIQA | 0.56 | 0.62 | 0.78 |
| Winogrande | 0.58 | 0.63 | 0.80 |
| CommonsenseQA | 0.58 | 0.65 | 0.72 |
| Social-IQA | 0.60 | 0.67 | 0.73 |
| ARC-Challenge | 0.61 | 0.70 | 0.81 |
| HellaSwag | 0.61 | 0.65 | 0.81 |
| MathQA | 0.61 | 0.67 | 0.76 |
| OpenBookQA | 0.61 | 0.64 | 0.78 |
| BoolQ | 0.61 | 0.68 | 0.80 |

## B.4 FEW-SHOT EVALUATION

We perform a 5-shot direct evaluation for all 61 models on 24 benchmarks to examine its impact (Gu et al., 2024) on cross-benchmark ranking agreement. The overall average Kendall's $\tau$ is 0.52 for direct evaluation (0-shot), 0.61 for direct evaluation (5-shot), and 0.76 for train-before-test (0-shot). Train-before-test outperforms 5-shot direct evaluation on 89% of benchmark pairs. We also present the mean ranking agreement between each benchmark and all others in Table 5. Train-before-test achieves better ranking agreement across all benchmarks. We view in-context learning as a weaker form of task preparation compared to fine-tuning—both give models task preparation, but fine-tuning is more thorough.

## B.5 IMPACT OF TEST SET SIZE

We experiment only with benchmarks with more than 2,000 test samples to understand how irreducible statistical noise (Fisher & Sen, 1994; Heineman et al., 2025) in test sets affects ranking agreement. The remaining benchmarks include MNLI, QQP, MedMCQA, QNLI, NQ-Open, HellaSwag, MathQA, BoolQ, HeadQA, and ARC-Easy. The overall average Kendall's $\tau$ across all benchmark pairs is 0.51 for direct evaluation (0-shot), 0.63 for direct evaluation (5-shot), and 0.80 for train-before-test. In other words, while the test set size impacts the statistical significance of the test scores, the cross-benchmark ranking agreement remains largely unchanged.

Table 6: The overall average Kendall's $\tau$ across all benchmark pairs for models in each size bin.

| Model size (B) | Direct evaluation | Train-before-test |
|---|---|---|
| $[0, 1)$ | 0.38 | 0.65 |
| $[1, 2)$ | 0.53 | 0.56 |
| $[2, 3)$ | 0.45 | 0.70 |
| $[6, 7)$ | 0.34 | 0.42 |
| $[7, 8)$ | 0.19 | 0.28 |
| $[9, 10)$ | 0.15 | 0.43 |
| All models | 0.52 | 0.76 |

## B.6 RANKING AGREEMENT FOR MODELS OF THE SAME SIZE

We group models into size bins, each containing at least 5 models, and compute the average Kendall's $\tau$ across all benchmark pairs for each bin (Table 6). Train-before-test consistently enhances ranking consistency compared to direct evaluation in every bin. The lower consistency observed when controlling for model size (compared to 0.76 for all models) is expected. Model potential strongly correlates with pretraining compute, as shown in Section 3.4, so removing size variation reduces the primary signal that distinguishes models.

Table 7: The overall average Kendall's $\tau$ across all benchmark pairs for each model family.

| Model family | Direct evaluation | Train-before-test |
|---|---|---|
| LLAMA | 0.51 | 0.65 |
| PYTHIA | 0.25 | 0.88 |
| QWEN | 0.57 | 0.79 |
| GEMMA | 0.43 | 0.77 |
| YI | 0.25 | 0.30 |
| All models | 0.52 | 0.76 |

Table 8: Explained variance ratios for the top five principal components of the score matrix for each model family, under direct evaluation and train-before-test, respectively.

| Model Family | Direct Evaluation | | | | | Train-Before-Test | | | | |
|---|---|---|---|---|---|---|---|---|---|---|
| | PC1 | PC2 | PC3 | PC4 | PC5 | PC1 | PC2 | PC3 | PC4 | PC5 |
| LLAMA | 69% | 21% | 4% | 3% | 2% | 90% | 4% | 3% | 1% | 1% |
| PYTHIA | 61% | 12% | 9% | 8% | 6% | 89% | 8% | 2% | 1% | 1% |
| QWEN | 74% | 9% | 5% | 3% | 3% | 93% | 2% | 1% | 1% | 1% |
| GEMMA | 58% | 27% | 6% | 4% | 2% | 92% | 3% | 2% | 2% | 1% |
| YI | 48% | 21% | 12% | 10% | 6% | 67% | 17% | 7% | 5% | 3% |
| ALL MODELS | 70% | 13% | 4% | 3% | 2% | 86% | 7% | 2% | 1% | 1% |

## B.7 RANKING AGREEMENT FOR MODELS OF THE SAME FAMILY

We group models by the model family, each containing at least 5 models, and compute the average Kendall's $\tau$ across all benchmark pairs for each family (Table 7). Train-before-test consistently achieves higher ranking consistency than direct evaluation for every family. The only anomaly occurs in the YI models, where ranking consistency by train-before-test is also quite low. This is possibly due to the pretraining compute for all considered YI-models being very similar; see Table 4.

We further conduct the PCA analysis in Section 3.4 for each model family, and report the explained variance ratios by the top five principal components. PC1 explains over 89% of the variance in four out of five families under the train-before-test scenario, exceeding 86% for all models. This suggests that differences among model families, including architectural choices and training data composition, may account for the remaining variance captured in higher-order principal components.

# C   ACCOUNTING FOR STATISTICAL SIGNIFICANCE

## C.1   RANKING ALIGNMENT IN FIGURE 1

We plot the rankings of 61 language models on two question-answering benchmarks: Natural Questions Open and ARC Challenge in Figure 1. We greedily align each ranking as much as possible without violating confidence intervals, thus revealing only those ranking changes that are statistically significant. See Algorithm 3 for more details.

## C.2   DOWNSTREAM RANKING AGREEMENT

We additionally supplement the experiments presented in the main text by modifying the ranking correlation metric to account for statistical significance in the benchmark evaluations. Specifically, we use Kendall's $\tau$-b (Kendall, 1945), which adjusts for ties in rankings. We consider two models tied on a given benchmark if their performance difference is not statistically significant at the 95% confidence level. We assess statistical significance using a t-test based on the standard error of the mean performances.

We reproduce the ranking correlation figures of the main text using the modified Kendall's $\tau$ which treats non-statistically significant performance differences as ties. See Figure 12 and 13; as well as Figure 14 and Figure 15 for more detailed results. We observe that accounting for statistical significance in models' performance differences leads to slightly higher ranking correlations, as measured by Kendall's $\tau$-b. For direct evaluation, average agreement increases from 0.52 to 0.58. For train-before-test, average agreement increases from 0.76 to 0.77. Therefore, train-before-test continues to lead to large improvements in raking agreement (from Kendall's $\tau$-b 0.58 to 0.77).

---

**Algorithm 1:** build_partial_order(scores, stderrs)

---

**Input:** Model performance scores and standard errors
**Output:** Directed graph $G$ representing significant model orderings
Initialize graph $G$ with models as nodes
**foreach** *pair of distinct models* $(m_1, m_2)$ **do**
  **if** $m_1$ *is significantly better than* $m_2$ **then**
  $\lfloor$ Add directed edge $(m_1 \rightarrow m_2)$ to $G$

**return** $G$

---

**Algorithm 2:** parallel_greedy_rank(models, $G_1$, $G_2$, score$_1$, score$_2$)

---

**Input:** List of models, two directed graphs $G_1$, $G_2$, and two score series
**Output:** Two lists representing the parallel ranking order for each task
Initialize `vanillaRank`$_1$, $\leftarrow$ rankdata(score$_1$), `vanillaRank`$_2$ $\leftarrow$ rankdata(score$_2$)
Initialize `available`$_1$ and `available`$_2$ as models with zero in-degree in $G_1$ and $G_2$
Initialize empty lists `order`$_1$, `order`$_2$
**for** $i = 1$ **to** *number of models* **do**
  Initialize empty list `pairs`
  **foreach** $m_1$ *in* `available`$_1$ **do**
    **foreach** $m_2$ *in* `available`$_2$ **do**
      Compute cost for pair $(m_1, m_2)$ based on:
        (1) Placement of $m_1$ in `order`$_2$ and $m_2$ in `order`$_1$
        (2) Whether $m_1 = m_2$ (prefer matching)
        (3) Combined vanilla ranks: `vanillaRank`$_2[m_1]$ + `vanillaRank`$_1[m_2]$
      Append $(cost, m_1, m_2)$ to `pairs`
  Sort `pairs` by cost (ascending)
  Select $(m_1, m_2)$ with minimal cost
  Append $m_1$ to `order`$_1$, $m_2$ to `order`$_2$
  Remove $m_1$ from $G_1$ and update `available`$_1$
  Remove $m_2$ from $G_2$ and update `available`$_2$
**return** `order`$_1$, `order`$_2$

---

**Algorithm 3:** rank_models(score$_1$, stderr$_1$, score$_2$, stderr$_2$)

---

**Input:** Scores and standard errors for two tasks
**Output:** Parallel rankings for both tasks
  $G_1 \leftarrow$ build_partial_order(score$_1$, stderr$_1$)
  $G_2 \leftarrow$ build_partial_order(score$_2$, stderr$_2$)
  $(order_1, order_2) \leftarrow$ parallel_greedy_rank(models, $G_1$, $G_2$, score$_1$, score$_2$)
  `rank`$_1[m]$ = position of $m$ in $order_1$ (starting from 1)
  `rank`$_2[m]$ = position of $m$ in $order_2$ (starting from 1)
**return** `rank`$_1$, `rank`$_2$

---

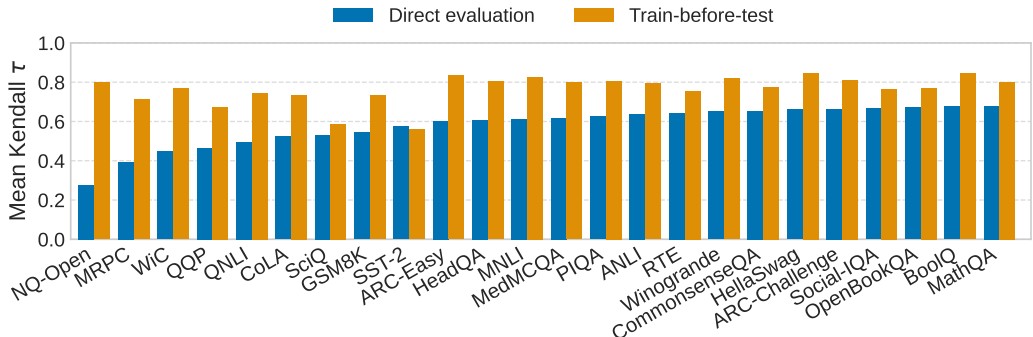

Figure 12: Mean ranking agreement between each benchmark and all others, measured by Kendall's $tau$-b, *with non-statistically significant performance differences being treated as ties*. We calculate Kendall's $\tau$-b between each benchmark and every other one, and then average. Compared to direct evaluation, train-before-test consistently improves ranking agreement–often by a large margin. On average, the overall average Kendall's $\tau$ is 0.58 for direct evaluation and 0.77 for train-before-test.

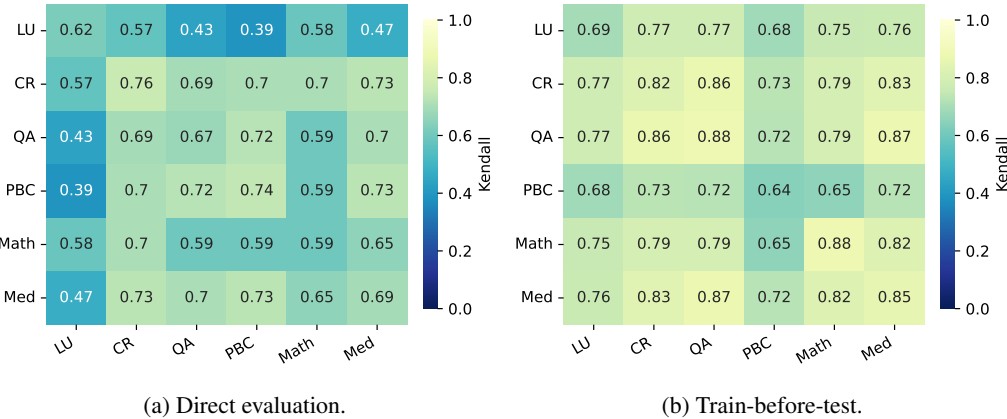

(a) Direct evaluation.

(b) Train-before-test.

Figure 13: Cross-category ranking agreement for direct evaluation (left) and train-before-test (right), measured by Kendall's $tau$-b, *with non-statistically significant performance differences being treated as ties*. We consider language understanding (LU), commonsense reasoning (CR), question answering (QA), physics/biology/chemistry (PBC), math (Math), and medicine (Med) categories. Kendall's $\tau$-b is averaged across all pairs of benchmarks that belong to two given categories. The diagonal represents the intra-category agreement and the others represent the inter-category agreement. train-before-test improves both intra- and inter-category ranking agreement in all instances.

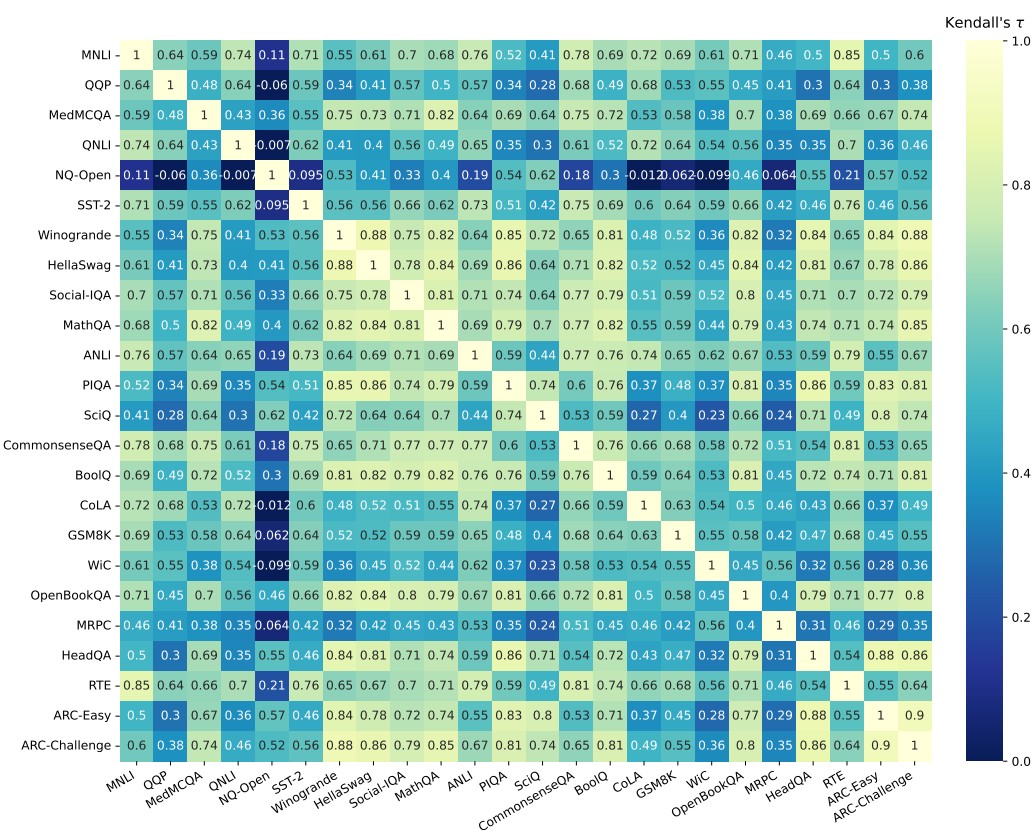

Figure 14: Cross benchmark ranking agreement under direct evaluation, measured by Kendall's $tau$-b with insignificant model comparisons treated as ties.

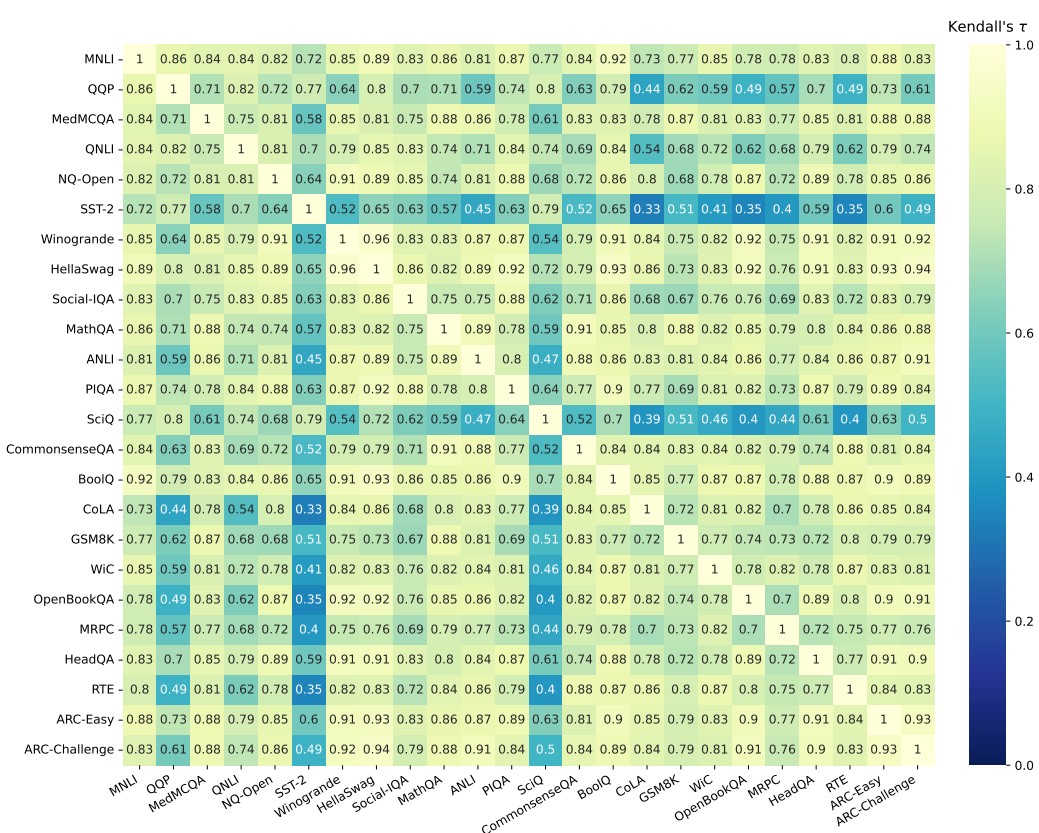

Figure 15: Cross benchmark ranking agreement under train-before-test, measured by Kendall's $tau$-b with insignificant model comparisons treated as ties.

## D    BROADER IMPACTS

We do not anticipate any direct societal impacts from this work, such as potential malicious or unintended uses, nor do we foresee any significant concerns involving fairness, privacy, or security considerations. Additionally, we have not identified potential harms resulting from the application of this technology.

## E    REPRODUCIBILITY STATEMENT

Detailed experimental settings are in Section 3.1 and Appendix A. Code is available at `https://github.com/socialfoundations/lm-harmony`.

## F    THE USE OF LARGE LANGUAGE MODELS

In this paper, we use large language models to aid and polish writing. Large language models are not used for retrieving related work or generating research ideas.

