# OpenReview forum: "Train-before-Test Harmonizes Language Model Rankings"
_ICLR.cc/2026/Conference — ICLR 2026 Oral_

### Official Review · Reviewer_WYQU · 2025-10-30

**Soundness:** 3
**Presentation:** 4
**Contribution:** 2
**Rating:** 6
**Confidence:** 4

**Summary:**

This paper studies how LLM performance varies across tasks. It proposes that instead of directly evaluating LLMs on downstream tasks, LLMs should first be finetuned on the training set of the downstream task (a paradigm they call "train-before-test"). The paper shows via a large-scale empirical evaluation that model rankings after train-before-test are much more correlated with each other across benchmarks than the out-of-the box performances are.

**Strengths:**

1. The paper provides a comprehensive empirical evaluation to show that a) model rankings change after finetuning and b) model ranks are much more correlated across benchmark tasks after finetuning.
2. The paper shows that pre-finetuning perplexity model rankings correlate with post-finetuning benchmark performance model rankings for base models but not for instruction-tuned models.
3. The paper shows that after fine-tuning, the model-benchmark score matrix becomes nearly rank 1.
4. The paper is quite well-written.

**Weaknesses:**

I see no major weaknesses of this paper. My biggest concern is its difference from [1], which is currently mentioned a few times in the paper. Lines 154-157 state that the paper introduces the "train-before-test" paradigm, but [1] already proposes to finetune on task-relevant data before evaluation (Section 2.1 of [1]). Can you more thoroughly articulate the differences between your paper and [1]? As it stands, I think this paper provides valuable insights about the stability of model rankings across tasks after finetuning, but the magnitude of this paper's contribution is smaller in light of [1].

[1]: "Training on the Test Task Confounds Evaluation and Emergence", Dominguez-Olmedo et al, ICLR 2025.

**Questions:**

1. "Model potential" does not feel like the right term for finetuned performance. To me, the natural meaning of "potential" would apply *before* finetuning instead of afterwards. The phrase is currently used to mean performance after finetuning, so I'd suggest thinking of a different term.

Small comments:
- Line 46: "\citet{Kaplan et al. (2020)}" -> "\citep{Kaplan et al. (2020)}"
- Line 52: I'd avoid "regret-free", since it's a technical term in online learning for sublinear regret, which is not what's being referred to here.
- Line 310: "\texttt{Wiki}pedia" -> "\texttt{Wikipedia}" and "\texttt{Stack}Exchange" -> "\texttt{StackExchange}"

---

> ### Author Response · Authors · 2025-11-21
>
> We appreciate the thoughtful feedback and for recognizing the value of our comprehensive empirical evaluation. We address your questions and suggestions below.
>
> **Difference from Dominguez-Olmedo et al. (2024).**
> Thank you for the question. While we build on the foundation established by Dominguez-Olmedo et al. (2024), our work addresses a fundamentally different research question with distinct empirical contributions.
>
> Dominguez-Olmedo et al. (2024) quantify the extent to which training on the test task confounds model evaluations. Their motivation is to understand why model families starting from late 2023 outperform prior models so strongly. They focus only on two benchmarks and do not provide any results for cross-benchmark agreement.
>
> In contrast, the very focus of our work is cross-benchmark agreement. We start from the observation that the benchmark ecosystem suffers from contradictory model rankings, with benchmarks testing similar skills producing different rankings, making model selection practically challenging (Figure 3). We perform comprehensive cross-benchmark agreement evaluations across 24 benchmarks. In doing so, we novelly establish train-before-test as a practical benchmarking methodology that harmonizes otherwise contradictory model rankings. Our work offers additional insights on the connection between perplexity and downstream benchmark performance (Section 3.3) and the latent factor behind model potential (Section 3.4). As Reviewer KmFP notes, *“While it is a follow-up to a previous ICLR paper, it adds net new contributions, notably showing how “train on test” fits into the existing language model benchmarking paradigm, which the previous work did not show explicitly.”*
>
> **Terminology about model potential.**
> Thank you for the suggestion. We use "model potential" to distinguish achievable performance after equal task-specific adaptation from out-of-the-box performance. This framing emphasizes that we're measuring what models can achieve rather than their current state—the relevant metric for practitioners selecting models for downstream tasks. Our perplexity experiments have shown that model potential is inherent to the models, not merely artifacts of fine-tuning (see Figure 5).
>
> We thank you for the insightful questions and helpful formatting suggestions. We have revised the paper to accommodate them.

---

### Official Review · Reviewer_KmFP · 2025-11-01

**Soundness:** 4
**Presentation:** 3
**Contribution:** 3
**Rating:** 8
**Confidence:** 4

**Summary:**

This work is follow-up to a previous ICLR 2025 paper proposing “training on the test set” as a better means of evaluating language models; this prior work showed that properties like “emergence” arise due to language models being tested on benchmarks that don’t take into account whether the model has been adapted to the downstream task; with “training on test set”, phenomena like emergence disappear. This work builds on this idea by showing further benefits of “training on test set”:

This work shows that after performing “training on test”, model rankings between similar benchmarks (e.g. benchmarks both testing QA) become more similar. Intuitively, one interprets this as – If you have multiple instantiations of benchmarks that all seem to test a common capability (e.g. medical knowledge vs math vs ..), then evaluating on those benchmarks in standard way can lead to very different evaluation results due to benchmark-specific properties (or if the models have trained in a manner arbitrarily to favor one benchmark but not the other), but with “train on test”, one is evaluating “potential” of the model on the underlying capability and not these other task-specific factors. In a similar analysis, this work shows that after performing “training on test”, model rankings between perplexity evaluations and downstream tasks are also more aligned.

**Strengths:**

This is a good paper and merits publication at ICLR. While it is a follow-up to a previous ICLR paper, it adds net new contributions, notably showing how “train on test” fits into the existing language model benchmarking paradigm, which the previous work did not show explicitly. This work also shows interesting finding about how to unlock correlation between perplexity and downstream benchmark, which has long been a frustrating topic in language modeling research. Experiments are sensible, varied in scope of models used in experimentation (though would’ve preferred to have seen more modern fully open models like Olmo or SmolLM; Pythia is outdated & not reproducible because not all data is public).

**Weaknesses:**

I think the work is good, so not too many critiques. I would say first, the heat map figures for rank correlation are hard to read (Fig 3, 4, 5). Maybe use a different color scheme, consider annotating where we should pay attention, and make caption a bit more self contained so we can read the Figure + grasp the finding instead of having to cross reference it with the body text.

Given the chosen models, I would’ve liked to have seen more findings that take advantage of: (1) comparing models of similar size across families, (2) comparing models within same family of different sizes. Aside from a Qwen analysis, this paper didn’t really dive into any of that, which is a shame.

**Questions:**

Even after this fine tuning, there is still not perfect correlation between (i) pairs of benchmarks measuring similar underlying capabilities, and (2) perplexity and downstream benchmarks. In fact, the best these rank correlations reach is somewhere around 0.7-0.8. That means there is something else irreducible that causes model rankings to be different. I would love to see authors provide a bit of discussion as to whether this is due to (a) intrinsic benchmark noise (an interesting reference would be Signal to Noise 2025 paper by Heineman et al though I understand it was published after this submission), (b) some differences in the underlying potential that these benchmark pairs are capturing, (c) some benchmark specific properties that even finetuning isn’t helping with, or something else? Just want to understand what is “left” between this work & maximal rank correlation.

Can you explain more the finding in Sec 3.4? In short, it seems that running PCA before train-then-test shows a single main principal component that is driving most of the variation in benchmark scores; and that this component is highly representative of pretraining compute. After train-then-test, it looks like, while the PC1 explains more variation, the conclusion about pretraining compute being the primary driver is still the same, is it not? I’m not understanding what net new learning we are getting here or how to interpret the significance of increasing the amount of variation that is explained by PC1.

Similarly, I’m not sure what is the takeaway from Qwen PC1’s analysis.

---

> ### Author Response · Authors · 2025-11-21
>
> We thank you for the thorough review and recommendation for acceptance. We address your questions and suggestions below:
>
> **Heat map readability.**
> We have updated Figures 3, 4, and 5 to enhance their readability and make them more self-contained in the revised version.
>
> **Model size and family analysis.**
> Thank you for this valuable suggestion. We have added two new analyses below.
>
> **Cross-family comparison (same size):** We grouped models into size bins with at least 5 models each and computed the average Kendall's $\tau$ across all benchmark pairs for each bin, reported in the table below.
>
> | Model size (B)   |   Direct evaluation |   Train-before-test |
> |:-----------------|--------------------:|--------------------:|
> | [0, 1)           |                0.38 |                0.65 |
> | [1, 2)           |                0.53 |                0.56 |
> | [2, 3)           |                0.45 |                0.7  |
> | [6, 7)           |                0.34 |                0.42 |
> | [7, 8)           |                0.19 |                0.28 |
> | [9, 10)          |                0.15 |                0.43 |
> | All models            |                0.52 |                0.76 |
>
>
> Train-before-test consistently improves ranking consistency compared to direct evaluation in every bin. The lower consistency observed when controlling for model size (compared to 0.76 across all models) is expected, as model potential strongly correlates with pretraining compute—removing size variation reduces the primary signal distinguishing models.
>
> **Within-family comparison (different sizes):** We analyzed model families with at least 5 models, as follows.
>
> | Model family   |   Direct evaluation |   Train-before-test |
> |:---------------|--------------------:|--------------------:|
> | Llama     |                0.51 |                0.65 |
> | Pythia     |                0.25 |                0.88 |
> | Qwen           |                0.57 |                0.79 |
> | Gemma         |                0.43 |                0.77 |
> | Yi          |                0.25 |                0.3  |
> | All models            |                0.52 |                0.76 |
>
>
> Again, train-before-test improves ranking consistency across all families. The only anomaly occurs in the 01-ai/Yi models, where ranking consistency by train-before-test is also quite low. This is possibly due to the pretraining compute for all considered Yi-models being very similar, with the largest difference being only 2x; see Table 4 in the Appendix.
>
> These results demonstrate that train-before-test improves ranking consistency both across and within model families. As expected, the effect is most pronounced when pretraining compute varies substantially.
>
> We have added these analyses to the Appendix of the revised manuscript.

---

> ### Author Response · Authors · 2025-11-21
>
> **Residual imperfect correlation.**
> This is an excellent question. The ranking disagreement between benchmarks is related to the rank of the score matrix. If the matrix were perfectly rank-1, all benchmarks would produce identical rankings (perfect correlation). The presence of higher-order principal components (PC2, PC3, etc.) means different benchmarks weight these components differently, leading to ranking disagreements. Therefore, understanding what PC2+ captures helps explain the residual imperfect correlation.
>
> When examining all 61 models, PC2+ explains 14% of the variance. However, when restricting to a single family (e.g., Qwen), PC2+ drops to only 7%. We further conducted PCA for each model family separately (see table below), which shows that PC1 explains more than 89% variance in four out of five families.
>
> Explained variance under train-before-test:
>
> |            |   PC1 |   PC2+ |
> |:-----------|------:|-------------------:|
> | Llama |    90% |     10% |
> | Pythia |    89% |     11% |
> | Qwen       |    93% |     7% |
> | Gemma     |    92% |     8% |
> | Yi      |    67% |    33% |
> | All models |    86% |    14% |
>
>
> This suggests that model family differences, including model architectural choices and training data composition, account for a substantial portion of ranking disagreement across benchmarks.
>
> Beyond family effects, the remaining imperfect correlation likely stems from: (1) measurement noise: As suggested by Heineman et al. (2025), benchmark scores contain inherent noise from finite test sets and stochastic evaluation procedures; (2) Incomplete adaptation: Our PEFT fine-tuning with light hyperparameter search may not fully unlock all model potential.
>
>
> **PCA analysis.**
> Thank you for highlighting this. The key insight is not simply that PC1 correlates with pretraining compute, but rather that train-before-test makes the score matrix nearly rank-1.
>
> Under direct evaluation, the score matrix has multiple meaningful dimensions (PC1: 70%, top 5: 91%). Under train-before-test, it becomes nearly rank-1 (PC1: 86%, suggesting a single dominant factor). This dramatic change indicates that train-before-test removes confounding factors, such as differential exposure to benchmark-related data during pretraining, that create artificial diversity in rankings. The fact that the matrix becomes nearly rank-1 after train-before-test validates our central claim: **model potential is fundamentally one-dimensional** (primarily determined by pretraining compute/scale), whereas out-of-the-box performance reflects multiple factors including task-specific preparation.
>
> Within the Qwen family, this effect is even more striking: PC1 explains 93% of variance under train-before-test, whereas PC2+ decreases from 14% (all models) to 7% (Qwen only), suggesting that the higher-order principal components in the full analysis largely capture model family differences.
>
> Thank you again for your thoughtful review and positive assessment of our work. We have updated the manuscript accordingly.

---

### Official Review · Reviewer_EhcJ · 2025-11-03

**Soundness:** 3
**Presentation:** 4
**Contribution:** 3
**Rating:** 8
**Confidence:** 5

**Summary:**

This work introduces a way compare model ranking based on performance of models after they are fine tuned on train split for a task. they say that this approach measures "model potential". The approach is dubbed "train-before-test". Through extensive experimental results, the paper shows that train-before-test stabilizes rankings of models: whereas before (under traditional protocol, which they dub "direct evaluation") rankings of models would change depending on benchmark, now rankings transfer between one model and the others. Overall, the paper represents an interesting contribution, and would benefit from being presented at ICLR.

**Strengths:**

- The paper is generally well motivated, easy to follow, and well written.
- Experimental setup is mostly sound (see weaknesses below), and the paper studies models across many LM families.
- I appreciate the framing of model potential as a mean to pick model to best adapt to a task. paper would benefit from stating the goal of train-before-test even more explicitly in the abstract: the presented technique is NOT an intrinsic evaluation of models as finished products, but as starting point for fine tuning.

**Weaknesses:**

- One limitation of this approach is that it does not provide an estimate of the magnitude of improvement. This could have been achieved by either proposing a way to average score, or use rankings to determine if any two models statistically different. Given the focus on practitioners using this method to choose models for downstream applications, providing a single, easy to interpret number is crucial.
- The paper lacks other comparison with other techniques that could be used to improve rankings. For example, the paper evaluates all models using shots, which gives puts models, especially those that are not instruction tuned at a major disadvantage [\(Gu et al., 2024\)](https://arxiv.org/abs/2406.08446). The test split of some of these benchmarks is also very small, and ranking stability would likely improve by adding training samples to the test set (see Fig 9 of [Heineman et al., 2025](https://arxiv.org/abs/2508.13144)) .
- I would caution against comparing with perplexity, as decontamination cannot be ensured for all models used.

**Questions:**

N/A

---

> ### Author Response · Authors · 2025-11-21
>
> We thank you for the positive assessment and valuable suggestions. We address your concerns below:
>
> **Magnitude of improvement.**
> While our primary focus is ranking consistency, we recognize the practical value of magnitude estimation. We calculated the average scores across 24 benchmarks under train-before-test, as shown in the Table below. Notably, PC1 scores (which capture model potential and correlate with pretraining compute per Section 3.4), align nearly perfectly with average scores (Kendall’s $\tau$=0.96). We also plotted the PC1 scores in Figure 11 in the Appendix.
>
>
> |    | model_name                          |   Avg |     PC1 |
> |---:|:------------------------------------|------:|--------:|
> |  1 | Qwen/Qwen2.5-14B                    | 0.784 |   5.959 |
> |  2 | Qwen/Qwen2.5-14B-Instruct           | 0.783 |   5.841 |
> |  3 | google/gemma-2-9b                   | 0.775 |   5.796 |
> |  4 | google/gemma-2-9b-it                | 0.771 |   5.437 |
> |  5 | google/gemma-7b                     | 0.752 |   4.615 |
> |  6 | Qwen/Qwen2.5-7B                     | 0.748 |   4.079 |
> |  7 | meta-llama/Meta-Llama-3-8B-Instruct | 0.747 |   4.239 |
> |  8 | Qwen/Qwen1.5-14B                    | 0.746 |   4.118 |
> |  9 | meta-llama/Llama-3.1-8B-Instruct    | 0.746 |   4.114 |
> | 10 | Qwen/Qwen2-7B-Instruct              | 0.744 |   3.996 |
> | 11 | Qwen/Qwen2-7B                       | 0.744 |   3.978 |
> | 12 | Qwen/Qwen2.5-7B-Instruct            | 0.744 |   3.794 |
> | 13 | meta-llama/Meta-Llama-3-8B          | 0.742 |   4.31  |
> | 14 | Qwen/Qwen1.5-14B-Chat               | 0.742 |   3.988 |
> | 15 | meta-llama/Llama-3.1-8B             | 0.742 |   4.217 |
> | 16 | 01-ai/Yi-1.5-9B-Chat                | 0.738 |   3.729 |
> | 17 | 01-ai/Yi-1.5-9B                     | 0.737 |   3.775 |
> | 18 | 01-ai/Yi-9B                         | 0.728 |   3.546 |
> | 19 | Qwen/Qwen1.5-7B                     | 0.72  |   2.797 |
> | 20 | 01-ai/Yi-1.5-6B-Chat                | 0.716 |   2.72  |
> | 21 | 01-ai/Yi-1.5-6B                     | 0.712 |   2.648 |
> | 22 | google/gemma-7b-it                  | 0.711 |   2.662 |
> | 23 | Qwen/Qwen2.5-3B                     | 0.71  |   2.243 |
> | 24 | Qwen/Qwen1.5-7B-Chat                | 0.71  |   2.429 |
> | 25 | meta-llama/Llama-3.2-3B-Instruct    | 0.707 |   2.362 |
> | 26 | Qwen/Qwen2.5-3B-Instruct            | 0.706 |   2.05  |
> | 27 | google/gemma-2-2b-it                | 0.701 |   2.155 |
> | 28 | 01-ai/Yi-6B                         | 0.699 |   2.168 |
> | 29 | 01-ai/Yi-6B-Chat                    | 0.697 |   2.176 |
> | 30 | meta-llama/Llama-3.2-3B             | 0.696 |   2.204 |
> | 31 | google/gemma-2-2b                   | 0.689 |   1.7   |
> | 32 | Qwen/Qwen1.5-4B                     | 0.689 |   1.487 |
> | 33 | Qwen/Qwen1.5-4B-Chat                | 0.683 |   1.23  |
> | 34 | Qwen/Qwen2.5-1.5B                   | 0.675 |   0.592 |
> | 35 | Qwen/Qwen2.5-1.5B-Instruct          | 0.668 |   0.316 |
> | 36 | Qwen/Qwen2-1.5B                     | 0.655 |  -0.172 |
> | 37 | Qwen/Qwen2-1.5B-Instruct            | 0.654 |  -0.209 |
> | 38 | google/gemma-2b                     | 0.641 |  -0.364 |
> | 39 | EleutherAI/pythia-12b               | 0.629 |  -0.843 |
> | 40 | Qwen/Qwen1.5-1.8B                   | 0.625 |  -1.506 |
> | 41 | Qwen/Qwen1.5-1.8B-Chat              | 0.62  |  -1.577 |
> | 42 | EleutherAI/pythia-6.9b-deduped      | 0.619 |  -1.233 |
> | 43 | meta-llama/Llama-3.2-1B-Instruct    | 0.619 |  -1.779 |
> | 44 | google/gemma-2b-it                  | 0.613 |  -1.644 |
> | 45 | meta-llama/Llama-3.2-1B             | 0.599 |  -2.354 |
> | 46 | Qwen/Qwen2.5-0.5B                   | 0.587 |  -3.348 |
> | 47 | Qwen/Qwen2.5-0.5B-Instruct          | 0.584 |  -3.42  |
> | 48 | EleutherAI/pythia-2.8b-deduped      | 0.58  |  -2.812 |
> | 49 | Qwen/Qwen2-0.5B-Instruct            | 0.573 |  -4.058 |
> | 50 | Qwen/Qwen2-0.5B                     | 0.572 |  -4.098 |
> | 51 | Qwen/Qwen1.5-0.5B                   | 0.566 |  -4.248 |
> | 52 | EleutherAI/pythia-1.4b-deduped      | 0.55  |  -4.235 |
> | 53 | Qwen/Qwen1.5-0.5B-Chat              | 0.548 |  -4.978 |
> | 54 | EleutherAI/pythia-1b-deduped        | 0.524 |  -5.73  |
> | 55 | openai-community/gpt2-xl            | 0.516 |  -5.762 |
> | 56 | openai-community/gpt2-large         | 0.491 |  -7.104 |
> | 57 | EleutherAI/pythia-410m-deduped      | 0.484 |  -7.495 |
> | 58 | openai-community/gpt2-medium        | 0.47  |  -8.469 |
> | 59 | EleutherAI/pythia-160m-deduped      | 0.44  | -10.639 |
> | 60 | openai-community/gpt2               | 0.431 | -11.421 |
> | 61 | EleutherAI/pythia-70m-deduped       | 0.401 | -13.968 |

---

> ### Author Response · Authors · 2025-11-21
>
> **Comparison with alternatives.**
> Thank you for the suggestion on the missing reference. We have added the discussion to our related work section. We show the additional experiment results as follows.
>
> **Few-shot evaluation:** We performed a 5-shot direct evaluation for all 61 models on 24 benchmarks. The overall Kendall’s $\tau$ is 0.52 for direct evaluation (0-shot), 0.61 for direct evaluation (5-shot), and 0.76 for train-before-test (0-shot). Therefore,  few-shot evaluation improves rank consistency, but train-before-test provides far greater consistency. We view in-context learning as a weaker form of task preparation compared to fine-tuning, but a form of preparation nonetheless.
>
> Here is the mean ranking agreement between each benchmark and all others (corresponding to Figure 2 in the paper):
>
> |               |   Direct evaluation (0 shot) |   Direct evaluation (5 shot) |   Train-before-test |
> |:--------------|-----------------------------:|-----------------------------:|--------------------:|
> | NQ-Open       |                         0.23 |                         0.44 |                0.74 |
> | MRPC          |                         0.34 |                         0.39 |                0.71 |
> | WiC           |                         0.39 |                         0.58 |                0.75 |
> | QNLI          |                         0.43 |                         0.58 |                0.74 |
> | QQP           |                         0.43 |                         0.62 |                0.77 |
> | CoLA          |                         0.45 |                         0.65 |                0.73 |
> | SciQ          |                         0.47 |                         0.49 |                0.71 |
> | SST-2         |                         0.5  |                         0.62 |                0.65 |
> | GSM8K         |                         0.51 |                         0.62 |                0.7  |
> | ANLI          |                         0.54 |                         0.56 |                0.78 |
> | MedMCQA       |                         0.55 |                         0.66 |                0.75 |
> | RTE           |                         0.55 |                         0.59 |                0.74 |
> | HeadQA        |                         0.55 |                         0.62 |                0.77 |
> | ARC-Easy      |                         0.55 |                         0.63 |                0.8  |
> | MNLI          |                         0.56 |                         0.63 |                0.8  |
> | PIQA          |                         0.56 |                         0.62 |                0.78 |
> | Winogrande    |                         0.58 |                         0.63 |                0.8  |
> | CommonsenseQA |                         0.58 |                         0.65 |                0.72 |
> | Social-IQA    |                         0.6  |                         0.67 |                0.73 |
> | ARC-Challenge |                         0.61 |                         0.7  |                0.81 |
> | HellaSwag     |                         0.61 |                         0.65 |                0.81 |
> | MathQA        |                         0.61 |                         0.67 |                0.76 |
> | OpenBookQA    |                         0.61 |                         0.64 |                0.78 |
> | BoolQ         |                         0.61 |                         0.68 |                0.8  |
>
>
> **Impact of test set size:** We further experimented only on benchmarks with more than 2000 test samples. The remaining benchmarks include MNLI, QQP, MedMCQA, QNLI, NQ-Open, HellaSwag, MathQA, BoolQ, HeadQA, and ARC-Easy. The overall Kendall’s $\tau$ for all three evaluation methods is 0.51 for direct evaluation (0-shot), 0.63 for direct evaluation (5-shot), and 0.80 for train-before-test. This suggests that train-before-test benefits are orthogonal to test set size—combining both would likely yield even better results.
>
> We have added these analyses to the Appendix of the revised manuscript.
>
> **Caution against perplexity comparison.**
> Thank you for the question. We designed our experiments to address the potential contamination problem. As stated at the beginning of Section 3.3, our perplexity corpora uses only 2025 data, and all evaluated models were released before 2025.
>
> We appreciate your recommendation for acceptance and hope these additions resolve your concerns.

---

### Official Review · Reviewer_ZDg6 · 2025-11-04

**Soundness:** 3
**Presentation:** 4
**Contribution:** 3
**Rating:** 6
**Confidence:** 5

**Summary:**

Disclosure: I reviewed an earlier version of this paper for NeurIPS. I had significant concerns with that draft, but the revised version is substantially improved.

The paper addresses a common problem in LLM benchmarking: benchmarks often result in contradictory model rankings, even if they are supposed to measure the same underlying capability (e.g., reasoning). The authors offer a new perspective on this problem: they suggest to focus on model potential after finetuning rather than model performance under direction evaluation. Thus, they propose to add a step of task-specific finetuning before evaluation, an approach they call _train-before-test_. In their experiments, the authors show that train-before-test leads to more consistent model rankings across benchmarks, and even increases the correlation with perplexity-based model rankings.

**Strengths:**

The problem addressed by the paper is important. The experiments conducted by the authors are very extensive, comprising a large set of LLMs and benchmarks. The paper is well written and easy to follow.

Compared with the NeurIPS submission, the revised framing in terms of model potential is persuasive and clarifies the significance of the experiments and results. I also liked the added sections connecting the work to the scaling-laws literature.

**Weaknesses:**

- The method proposed by the authors only works in the narrow setting of LLM evaluation _where subsequent task-specific finetuning is guaranteed_. As a result, train-before-test provides little evidence about performance in typical deployment without such fine-tuning. The authors note this in the discussion, but because deployment without task-specific fine-tuning is far more common, this is a substantial limitation and should be stated upfront &mdash; in both the abstract and the introduction &mdash; to avoid readers over-generalizing the scope of the paper.

- The central results of the paper are expected given established facts about the connection between (i) model size/pretraining compute and (ii) downstream performance after task-specific finetuning. The main empirical take-away (supported by the PCA analysis in section 3.4) is that larger/more pretrained models are consistently better across downstream tasks than smaller/less pretrained ones _after_ task-specific finetuning, but not _before_. However, this appears to follow directly from prior work:
  - After task-specific finetuning, model size/pretraining compute (in combination with task-specific features) strongly predict downstream task performance (e.g., [Lin et al., 2024](https://arxiv.org/abs/2402.02314); [Zhang et al., 2024](https://arxiv.org/abs/2402.17193)).
  - Under direct evaluation (i.e., without finetuning), model size/pretraining compute are much less predictive of downstream task performance (e.g., [Magnusson et al., 2024](https://arxiv.org/abs/2312.10523); [Lourie et al., 2025](https://arxiv.org/abs/2507.00885)).

   The paper would be stronger if this connection were made explicit. Given that the authors already refer to the scaling-laws literature, adding this would be relatively easy.

**Questions:**

You write in lines 97 to 99: "Perplexity benchmarks used to be popular, but fell out of fashion because of the apparent disconnect between perplexity and downstream task performance" &mdash; can you clarify what specifically you base this claim on? In contemporary LLM development, perplexity remains a primary evaluation metric, despite all its limitations.

---

> ### Author Response · Authors · 2025-11-21
>
> Thank you for your thorough evaluation and constructive feedback. Below, we address the concerns you raised.
>
> **Scope and applicability of our work.**
> We thank you for raising this important point about scope. Following your recommendation, we have revised both the abstract and introduction to explicitly state that train-before-test is designed for scenarios where task-specific finetuning will be performed, and it provides a complementary lens to direct evaluation, which remains essential for assessing out-of-the-box deployment.
>
> Within this more clearly defined scope, we believe the contribution remains significant. LLM adaptation has a substantial and growing presence both in industry and academia. In practice, major LLM providers (OpenAI, Anthropic, Google) now offer finetuning APIs as core services, and adaptation is routinely employed in domains like healthcare, legal, and enterprise applications. In research, there is extensive work on LLM adaptation. For example, the papers you cited (Lin et al., 2024; Zhang et al., 2024) specifically focus on finetuning scenarios, indicating this is a very significant and active research direction.
>
> Beyond practical concerns, our work also reveals fundamental insights about model potential. Specifically, our perplexity experiments (Section 3.3, Figure 5) demonstrate that model potential is an inherent property of base models, not an artifact of finetuning: pre-finetuning perplexity on held-out data strongly predicts post-finetuning downstream performance. This suggests a stable property of base models that exists prior to any task-specific adaptation.
>
> We appreciate your feedback, which has substantially improved the clarity of our contribution.
>
>
> **Novelty of our empirical findings.**
> We appreciate this suggestion and have added more explicit discussion about the scaling laws literature in the Related Work section. However, we respectfully disagree that our main results are "expected given established facts."
>
> You cite Lin et al. (2024) and Zhang et al. (2024) as establishing that model size/pretraining compute predicts downstream performance after finetuning. While these papers make valuable contributions, neither study cross-benchmark ranking agreement. Lin et al. (2024) predict full-finetuning performance of a single model from partial finetuning on one task using their rectified scaling law. Zhang et al. (2024) study how different factors scale within individual tasks. Notably, their scaling laws are model-specific and task-dependent, and thus they say little about what to expect regarding cross-benchmark rank agreement.
>
> Our central contribution (that finetuning harmonizes rankings across benchmarks) is not addressed in prior work. Our empirical findings on the predictability of post-finetuning performance from pre-finetuning perplexity (Section 3.3) and on how fine-tuning renders the score matrix essentially rank-1 (Section 3.4) are also not established in the literature.
>
> We have revised the Related Work section to draw these distinctions more explicitly.
>
> **On the perplexity claim.**
> We appreciate your request for clarification. You correctly note that perplexity remains essential for monitoring training progress during LLM development. Our statement refers specifically to its diminished role in public benchmarking and comparative model evaluation, not its use during development itself.
>
>
> This distinction is documented in the literature you cite: Lourie et al. (2025) explicitly states “Better perplexity does not always translate to better downstream performance; perplexity is not all you need.” This disconnect motivated the field's shift toward task-specific benchmarks for public model comparison, as evidenced by prominent leaderboards (e.g., HELM, OpenLLM Leaderboard) emphasizing downstream tasks over perplexity.
>
> We thank you for the question and have refined this statement to be more precise in the revised version.

---

### Author Response · Authors · 2025-11-25

We sincerely thank all reviewers for their thorough evaluations and constructive feedback. The feedback has significantly strengthened our paper, and we have carefully addressed all concerns raised.

In this response, we have marked all substantive revisions in blue. Below is a summary of the key changes made to the original manuscript:

- Revised abstract and introduction to clarify the scope of train-before-test (Reviewers ZDg6, EhcJ).
- Revised related work:
- - To discuss connections with scaling laws literature (Lin et al., 2024; Zhang et al., 2024a) (Reviewer ZDg6)
- - To clarify novelty beyond Dominguez-Olmedo et al. (2024) (Reviewer WYQU)
- - To add related works on standard evaluation practices (Gu et al., 2024; Heineman et al., 2025) (Reviewers EhcJ, KmFP)
- Added experimental results in Appendix B.4-B.7:
- - Few-shot evaluation comparison (Reviewer EhcJ)
- - Test set size impact analysis (Reviewer KmFP)
- - Same-size and same family model ranking analyses (Reviewer KmFP)
- Added interpretations and clarifications:
- - Additional interpretation on PCA results (Reviewer KmFP)
- - Discussion on imperfect ranking correlation (Reviewer KmFP)
- - Clarification on perplexity usage in public benchmarking (Reviewer ZDg6)
- Enhanced presentation:
- - Improved figure readability with more self-contained captions (Reviewer KmFP)

Additionally, we made minor improvements to writing clarity, formatting, and presentation throughout the manuscript. We also included the appendix in the manuscript, which was previously in the supplement.

We believe these revisions have addressed the concerns raised and significantly improved the paper's quality. We thank the reviewers for their constructive feedback.

---

### Meta-Review · Area_Chair_jEZh · 2025-12-21

**Summary:**

While a large body of studies evaluate LLMs, researchers observe contradictory model rankings across benchmarks, which hinder model selection and comparison. This paper addresses a critical challenge in robustly evaluating and ranking LLMs’ performance. The proposed train-before-test standardizes benchmark-specific fine-tuning for all models prior to evaluation, aiming to measure "model potential" rather than conventional out-of-the-box deployment performance.

The reviewers raise several core concerns: (1) novelty relative to prior evaluation/fine-tuning work, (2) robustness across model sizes/benchmark types, (3) methodological clarity (e.g., perplexity correlation), and (4) over-generalization/over-claim on the scope.

The authors’s rebuttal have duly addressed the main concerns and made substantive revisions accordingly. Minor outstanding concerns are non-critical for the paper’s significant contributions to LLM evaluation.

In summary, given its novelty and wide significance of this work for LLM research, I recommend the submission for Accept (Oral).

**Reviewer Concerns:**

I think the authors have addressed nearly all core concerns. As the summary made by the authors suggest, the revised contents addressed the concerns accordingly.

Addressed Main Concerns:
- Revised abstract and introduction to clarify the scope of train-before-test (Reviewers ZDg6, EhcJ).
- Revised related work on scaling laws literature (Reviewer ZDg6), novelty beyond Dominguez-Olmedo et al. (2024) (Reviewer WYQU),  standard evaluation practices. (Reviewers EhcJ, KmFP).
- More experimental results in Appendix B.4-B.7, including Few-shot evaluation comparison (Reviewer EhcJ), Test set size impact analysis (Reviewer KmFP), and Same-size and same family model ranking analyses (Reviewer KmFP)
- More interpretations and clarifications: Additional interpretation on PCA results (Reviewer KmFP), Discussion on imperfect ranking correlation (Reviewer KmFP), Clarification on perplexity usage in public benchmarking (Reviewer ZDg6).

I do not see significant outstanding weakness.

**Reviewer Scores:**

- Reviewer ZDg6: 6 -> 8

Justification: I think the authors have addressed the raised main concerns.

- Reviewer EhcJ: 8 -> 8

Justification: I think the authors have addressed the mentioned minor concerns with additional experiments, while the response to W1 is not fully convincible.

- Reviewer KmFP: 8 -> 8

Justification: I think the authors have addressed the mentioned minor concerns with additional experiments and revisied contents (Model size and family analysis).

- Reviewer WYQU: 6 -> 8

Justification: I think the authors have addressed the only one main concern on relation to previous literature.

---

### Decision · Program_Chairs · 2026-01-26

Accept (Oral)